

# Characterization of nitrous acid and its potential effects on secondary pollution in warm-season of Beijing urban areas

Junling Li[1], Chaofan Lian[2,#], Mingyuan Liu[3], Hao Zhang[1], Yongxin Yan[1], Yufei Song[1], Chun Chen[1], Haijie Zhang[1], Yanqin Ren[1], Yucong Guo[2], Weigang Wang[2], Yisheng Xu[1], Hong Li[1,*], Jian Gao[1,*], Maofa Ge[2,*]

[1] State Key Laboratory of Environmental Criteria and Risk Assessment, Chinese Research Academy of Environmental Sciences, Beijing 100012, China
[2] State Key Laboratory for Structural Chemistry of Unstable and Stable Species, Beijing National Laboratory for Molecular Sciences (BNLMS), CAS Research/Education Center for Excellence in Molecular Sciences, Institute of Chemistry, Chinese Academy of Sciences, Beijing 100190, China
[3] China National Environmental Monitoring Centre, Beijing, 100012, China
[#] Now at Assessment and Research Center for Pollution and Carbon Reduction, Tianfu Yongxing Laboratory, Chengdu 610213, China

*Correspondence to*: Hong Li (lihong@craes.org.cn), Jian Gao (gaojian@craes.org.cn), Maofa Ge (gemaofa@iccas.ac.cn)

**Abstract.** Benefiting from a series of pollution prevention initiatives, fine particle ($PM_{2.5}$) pollution was effectively controlled in 2018-2020, but ground-level ozone pollution during warm season has become a major issue for the continuous air quality improvement in Beijing. As a key source of hydroxyl (OH) radical, nitrous acid (HONO) has attracted much attention for its important role in the atmospheric oxidant capacity (AOC) increase; the elucidation of the pollution characteristics, unknown sources and the contribution to secondary pollution of HONO has become a research hotspot. In this study, we made a comparative study on the ambient levels, variation patterns, sources and formation pathway in warm season (from June to October in 2021) on a basis of a continuous intensive observation in an urban site of Beijing. The monthly average mixing ratio of HONO were 1.26 ppb, 1.28 ppb, 1.01 ppb, 0.96 ppb, 0.89 ppb, respectively, showing a larger contribution to OH radical relative to ozone at daytime with a mean OH production rate of 2.70, 2.91, 2.00, 2.25, and 0.93 ppb/h, respectively. The emission factor from the vehicle emissions, $P_{emis}$, was estimated to be 0.017, higher than most studies conducted in Beijing, the observation site and traffic control policies could affect this phenomenon. The homogeneous production of HONO via reaction of NO + OH, $P_{netOH+NO}$, in each month were 0.050, 0.045, 0.033, 0.052, and 0.17 ppb/h, respectively. The average nocturnal $NO_2$ to HONO conversion frequency $C_{HONO}$ in each month were 0.011 $h^{-1}$, 0.0096 $h^{-1}$, 0.013% $h^{-1}$, 0.0081 $h^{-1}$, and 0.0017 $h^{-1}$, respectively.

In warm seasons, the missing source of HONO, $P_{unknown}$, around noontime were 0.29-2.37 ppb/hr. $P_{unknown}$ in each month might be various. According to the observation results, relatively low humidity and strong solar illumination were conductive to HONO formation in June, which might be due to light-induced heterogeneous reactions of $NO_2$. In July in Beijing, high humidity condition was beneficial to the heterogeneous reaction of $NO_2$, and due to the increase of precipitation, more HONO would enter the liquid phase (the high Henry coefficient of HONO). For days with high humidity and strong sunlight in June and August, photolysis of nitrate was also one important HONO source. For August and September, light-induced reactions of $NO_2$ on non-aerosol surfaces under relatively low humidity and strong light conditions could be an important HONO source. In addition, the presence of Cl ions and sulfate could enhance the photolysis of nitrate, and this was obvious in July and October; the presence of organic compounds also could have this effect, which was obvious



in June and October. Not only the HONO concentration but also the HONO source has temporal patterns, even within a

season, it varies from month to month. This work highlights the importance of HONO for AOC in warm season, while encouraging long-term HONO observation to assess the contribution of HONO sources over time compared to the capture of pollution processes.

## 1 Introduction

From 2013 to 2022, the annual average concentration of $PM_{2.5}$ in China has been decreasing, but it is still much higher

than the World Health Organization's guideline value (5 μg/m$^3$). Ozone pollution, on the other hand, is becoming more and more prominent (Liu et al., 2023b;Li et al., 2022). China's air quality problem has developed from being dominated by $PM_{2.5}$ in the past to being affected by the photochemical pollutants $PM_{2.5}$ and ozone, that is, "Air Pollution Complex" (Zhu et al., 2023). The atmospheric oxidation capacity (AOC) determines the production rates of secondary pollutants in the atmosphere, and is the essential driving force in forming complex air pollution (Liu et al., 2021b).

As a trace gaseous pollutant, nitrous acid (HONO) strongly impacts the AOC, the photolysis of which can contribute to more than 60% of the OH radicals at daytime (Czader et al., 2012). As an important source of OH radicals in both rural and urban environments, HONO outweighs the contribution of ozone photolysis (Gu et al., 2022;Elshorbany et al., 2012;Yang et al., 2021b). Thus, as an important source of oxidants for AOC, HONO can substantially influence the formation of secondary pollutants, exerting a considerable impact on air quality, climate change, and human health (Requia et al.,

2018;Zhang et al., 2021b;Huang et al., 2018;Liu et al., 2023b).

Due to the importance of HONO to AOC, researchers have done lots of work on the transformation process of HONO, i.e., source and sink. For the source of HONO, it is complicated and has been debated for decades. To sum it up, source of atmospheric HONO mainly includes: (a) direct emission from combustion, soil, and livestock farming. The combustion includes vehicle exhaust, industrial exhaust, and biomass burning (Liao et al., 2021;Kurtenbach et al., 2001;Kirchstetter et al.,

1996;Nie et al., 2015). The release from soil nitrite is one important primary source of HONO (Su et al., 2011;Wu et al., 2019), the biological soil crusts can accelerate the HONO emission in drylands (Weber et al., 2015), fertilization behavior of agricultural fields remarkably enhance the emission of HONO (Xue et al., 2021), and denitrification is one major HONO production pathway in boreal agricultural fields (Bhattarai et al., 2021). Meanwhile, livestock farming is also identified as one important primary emission source of HONO (Zhang et al., 2023a). (b) Homogeneous reaction of OH +NO, which

usually occurs in polluted areas, the contribution of this recombination pathway to HONO may be significant during daytime with high NO and OH concentration (Gu et al., 2022). (c) Heterogeneous conversion of $NO_2$ on various surfaces (Liu et al., 2023a;Yu et al., 2022a;Yang et al., 2021a), Finlayson-Pitts et al. (2003) reported that $NO_2$ could be converted to HONO on humid surfaces with first order in $NO_2$, Xu et al. (2015) highlighted the HONO source had a large variability with heterogeneous conversion of $NO_2$ at ground face, Aubin and Abbatt (2007) found that the heterogeneous conversion of $NO_2$

on hydrocarbon surfaces was an effective pathway to generate HONO. (d) photolysis of nitric acid and nitrate in particles at





daytime, which were considered as important sources of HONO in both experiments and field observations (Wang et al., 2023b;Ye et al., 2019;Yang et al., 2018;Ye et al., 2017). And for the sinks of HONO, diurnal photolysis is the main consumption way (Keller-Rudek et al., 2013), the reaction of HONO with OH radical is also one sink that needs to be paid attention to (Cox et al., 1976). In addition, HONO can also be removed by dry deposition, and the removal rate is related to
the molecular free path and the mixed layer height (Kessler and Platt, 1984).

Generally, HONO concentration in the atmosphere is closely related to the quality of the atmosphere. As the heterogeneous processes are considered as the main source of HONO, and most of which are poorly understood, therefore, many field observations of HONO were focus on haze polluted environments (Li et al., 2012;Spataro et al., 2013;Tong et al., 2015;Xu et al., 2015;Hou et al., 2016). In urban areas, the HONO concentration can be up to more than 10 ppb during severe
pollution episode (Zhang et al., 2019b). In China, the Chinese government has implemented series of stringent clean air actions to address the severe air pollution issue since 2013, significant reduction of $PM_{2.5}$ was observed in North and South China during 2013-2020 (Li et al., 2022). However, ozone levels have not been effectively reduced in China, the warm-season (from April to September) mean maximum daily 8 h average ozone (MDA8 $O_3$) increased by 2.6 µg/(m$^3$·yr) (Liu et al., 2023b). In view of the atmospheric changes in $PM_{2.5}$ and ozone, recent field observations have begun to pay attention to
the potential influence of HONO on $PM_{2.5}$, ozone, and coexistence of $PM_{2.5}$ and ozone (Zhang et al., 2023c;Zhang et al., 2023b;Xuan et al., 2023;Chen et al., 2023;Zhang et al., 2022;Lin et al., 2022;Li et al., 2021). However, most HONO observations were conducted in autumn and winter, and the period were relatively short, usually a few weeks covering one or several pollution processes. Based on the increase of ozone in warm-season, long-term observations during the warm season are needed, yet is rather limited.

In this work, we conducted one HONO observation in an urban site of Beijing, the time was from June to October 2021, covering the whole summer and part of autumn. According to the Technical Regulation on Ambient Air Quality Index in China, only one day of moderate pollution ($PM_{2.5}$, in October) and no more than four days of light pollution (ozone) per month were included, so we analysed the observation data on a monthly basis. The atmospheric levels and variations of HONO and related species during each month were analysed and compared. The impacts of HONO on atmospheric oxidant
capacity in each month were also assessed.



## 2 Methods

### 2.1 Observation site and instruments

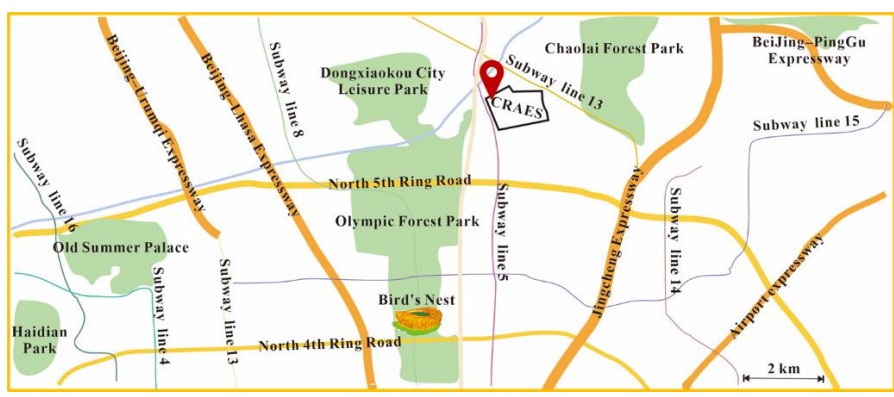

**Figure 1: Location of the observation site in this work.**

The observation campaign was conducted from June 18th, 2021 to October 25th, 2021. The observation site was located at Chinese Research Academy of Environmental Sciences (CRAES, 40°04´N, 116°42´E), an urban site (Ren et al., 2022;Zhang et al., 2021a) in Beijing, China (Figure 1).

HONO was measured with a water-based long-path absorption photometer (Chen et al., 2020), the potential interferences, e.g., hydrolysis of $NO_2$, was subtracted by deployed a dual-channel absorption system. A set of commercial
analyzers (Thermo 42i, 43i, 48i, 49i, 5030i, USA) were used to measure the concentrations of $NO_x$, $SO_2$, CO, $O_3$, and $PM_{2.5}$ online. Due to the employment of a molybdenum $NO_2$-to-NO converter, the 42i analyzer might overestimate $NO_2$ concentration for the potential conversation of $NO_z$ ($NO_z = NO_y - NO_x$. e.g., HONO, $HNO_3$, peroxyacetyl nitrate (PAN), and so on). Considering the relatively lower concentration of $NO_z$ compared with $NO_2$, this impact would be minor (Zhang et al., 2022). The time resolution of the analyzers above were 1min, and the detection limits were 0.4 ppbv, 0.5 ppbv, 0.04 ppmv,
1ppbv, and 0.5 μg/m$^3$, respectively. In order to ensure the accuracy of the data, routine maintenance was carried out. The meteorological parameters (temperature, T; relative humidity, RH; pressure, P; wind speed and direction, WS and WD) were obtained with an automatic weather station (MAWS301, Vaisala, Finland) with a time resolution of 1 hr. Mass concentrations of the inorganic compositions in $PM_{2.5}$ ($NO_3^-$, $SO_4^{2-}$, $Cl^-$) were detected with a Monitor for AeRosols and Gases in ambient Air (MARGA, Model ADI 2080, Applikon Analytical B.V., the Netherlands) with 1 hr resolution. Detailed
descriptions of the instrument inlet design and the operating characteristics can be referred in previous studies (Wang et al., 2022).

### 2.2 Photolysis rate simulation and OH estimation

Photolysis frequencies of $JNO_2$ was measured by a filter radiometer (METCON, Germany) with a time resolution of 1 min. Photolysis rate constant of $O_3$, HONO, and other parameters were simulated according to the aerosol optical depth and solar





zenith angle by the weather research and forecasting model coupled with chemistry (WRF-Chem, version 3.7.1), based on the TUV model (http://cprm.acom.ucar.edu/Models/TUV/InteractiveTUV/) (Zhang et al., 2019a). To reduce the imprecision of model simulation, the observed $JNO_2$ value was used to correct simulated results, i.e., $J(O^1D)$ and $J(HONO)$.

The OH concentration was calculated with the following equation (Liu et al., 2019):

$$[OH] = a \times (\frac{J(O^1D)}{10^{-5}s^{-1}})^b + c \qquad (1)$$

where a = 4.2 × $10^6$ cm$^{-3}$, b = 1, c= 2.2 × $10^5$ cm$^{-3}$ (Lin et al., 2022). At night, $J(O^1D)$ was zero, the nighttime OH concentration here was 2.2 × $10^5$ cm$^{-3}$. On the daytime, the OH concentration was estimated based on the equation above.

## 3 Results and discussion

### 3.1 Observation data overview

#### 3.1.1 Variations of meteorological parameters during observation period

Figure S1 shows the time series of meteorological parameters including RH, T, WS, WD, and $JNO_2$ from June 18th, 2021 to October 25th, 2021. During the observations, the temperature ranged from 3.10-37.27 ℃ with an average of 22.99 ℃, and the RH ranged from 7.08%-100% with an average of 59.44%. The average wind speed during the observation was 1.11 m/sec, and the maximum wind speed was 5.20 m/sec. The average and monthly average values of T, RH, and WS are shown in Table S1. As shown in Figure S2 and Table S1, the temperature in June, July, and August was higher, 25.98-28.38 ℃,

temperature begun to drop slightly in September, 22.33 ℃, in October, temperature dropped significantly, 13.42 ℃. Meanwhile, the solar radiation in September and October decreased as the sun moved south. The maximum value of $J(NO_2)$ during the observation was 8.21×10$^{-3}$ s$^{-1}$, the values reached a maximum at noon (with high solar radiation) and decreased to zero at night. The average wind speed also slightly decreased in September and October. During the observation, June had the lowest RH and the maximum wind speed; and due to the increase in precipitation, July had the highest RH. For the daily

variation of meteorological parameters, T and $JNO_2$ expressed the same variation pattern with $O_3$, that was increasing after sunrise and decreasing after sunset, as shown in Figure S2. While the RH showed the opposite pattern that was increasing during night and decreasing during the daytime. For the WS, it was higher from 8 to 20 o'clock in June, and for the rest of the observation period, WS begun to increase in the afternoon and subside after sunset.

#### 3.1.2 Variations of pollutant species during observation period

Figure 2 shows the time series of basic parameters including HONO, $NO_2$, NO, CO, $O_3$, $SO_2$, $PM_{10}$, $PM_{2.5}$, and $HONO/NO_2$ from June 18th, 2021 to October 25th, 2021. The HONO concentration ranged from 0.05 ppb to 5.17 ppb, with an averaged value of 1.06 ppb. The concentration of $O_3$ ranged from 1ppb to 227 ppb, with an averaged value of 71.58 ppb. The concentrations of $NO_2$, NO, CO, $SO_2$, and $PM_{2.5}$ were 1-113 ppb, 0.01-209.58 ppb, 0.1-1.6 ppm, 1.34-4.56 ppb, and 1-181 µg/m$^3$, respectively, with average values of 25.67 ppb, 7.87 ppb, 0.56 ppm, 1.89 ppb, 19.43 µg/m$^3$, respectively. The average





and monthly average concentrations of the basic parameters are shown in Table 1. One haze episode (1 day) occurred across the observation period, the daily mass concentration of $PM_{2.5}$ was 117 μg/m³, higher than the National Ambient Air Quality Standard (Class II: 75 μg/m³). And for $O_3$, eight episodes (13 days) occurred across the observation period, with MDA8 $O_3$ higher than the Class II: 160 μg/m³. Overall, the high HONO concentration was always accompanied by high concentration of $NO_x$ (NO, $NO_2$) or aerosols ($PM_{2.5}$).

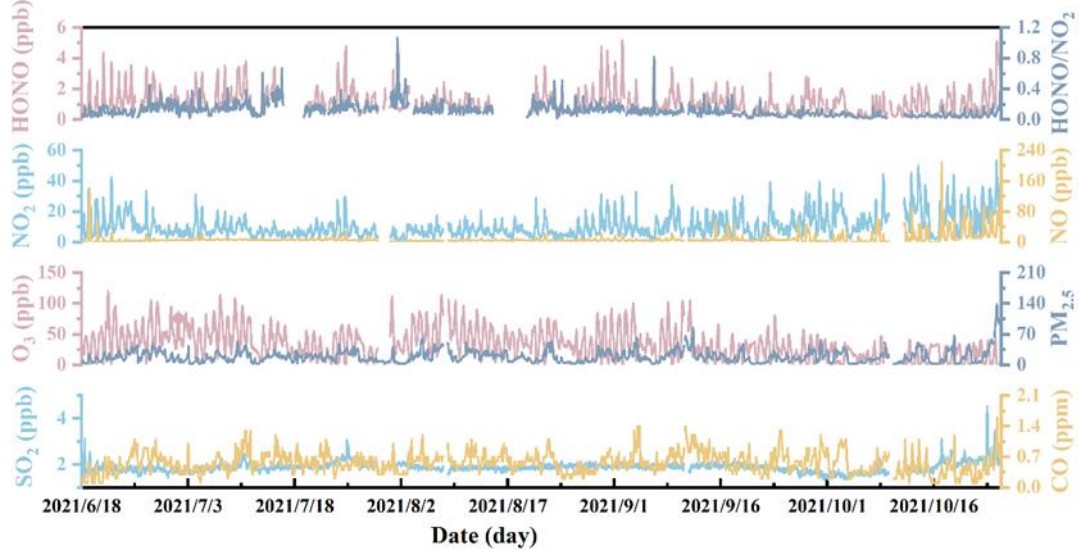


**Figure 2: Temporal variations of hourly average $PM_{2.5}$, CO, $O_3$, $SO_2$, NO, HONO, $NO_2$, and HONO/$NO_2$ during the observation period.**

**Table 1. The average and monthly average concentration of HONO, $NO_2$, NO, CO, $O_3$, $SO_2$, $PM_{2.5}$, HONO/$NO_2$, and HONO/$NO_2$ from June to October during the observation period.**

| concentration | | HONO (ppb) | NO₂ (ppb) | HONO/NO₂ | NO (ppb) | NOₓ (ppb) | HONO/NOₓ | CO (ppm) | O₃ (ppb) | SO₂ (ppb) | PM₂.₅ (μg/m³) |
|---|---|---|---|---|---|---|---|---|---|---|---|
| average | | 1.06 | 25.67 | 0.052 | 7.87 | 33.01 | 0.040 | 0.56 | 71.58 | 1.89 | 19.43 |
| Jun. | average | 1.26 | 28.61 | 0.051 | 5.72 | 34.24 | 0.044 | 0.52 | 87.56 | 1.81 | 20.11 |
| | max | 4.40 | 92 | 0.17 | 139.22 | 217.22 | 0.56 | 1.1 | 227 | 4 | 51 |
| | min | 0.14 | 9 | 0.0054 | 0.3 | 1.7 | 0.0049 | 0.1 | 1 | 1.41 | 3 |
| Jul. | average | 1.28 | 19.88 | 0.072 | 6.55 | 25.96 | 0.052 | 0.56 | 78.35 | 1.93 | 16.63 |
| | max | 4.78 | 81 | 0.55 | 28.53 | 92.58 | 0.26 | 1.3 | 211 | 3.07 | 53 |
| | min | 0.11 | 2 | 0.0061 | 2.17 | 3.8 | 0.0051 | 0.1 | 1 | 1.51 | 1 |
| Aug. | average | 1.01 | 18.17 | 0.079 | 4.80 | 22.5 | 0.055 | 0.57 | 87.04 | 1.89 | 18.88 |
| | max | 4.77 | 79 | 0.79 | 24.47 | 87.45 | 0.31 | 1.2 | 214 | 2.21 | 60 |
| | min | 0.052 | 1 | 0.0049 | 1.3 | 2.52 | 0.0039 | 0.2 | 10 | 1.54 | 2 |
| Sep. | average | 0.96 | 29.56 | 0.035 | 6.730 | 36.04 | 0.028 | 0.65 | 68.11 | 1.87 | 19.44 |
| | max | 5.17 | 84 | 0.19 | 57.45 | 115.71 | 0.17 | 1.4 | 210 | 2.26 | 86 |
| | min | 0.078 | 3 | 0.0042 | 0.39 | 3.5 | 0.0037 | 0.2 | 9 | 1.34 | 1 |
| Oct. | average | 0.89 | 36.01 | 0.025 | 16.60 | 50.91 | 0.021 | 0.46 | 39.54 | 1.92 | 23.30 |
| | max | 5.1 | 113 | 0.21 | 209.58 | 252.58 | 0.53 | 1.6 | 116 | 4.56 | 181 |
| | min | 0.066 | 1 | 0.0041 | 0.01 | 1.78 | 0.0022 | 0.1 | 10 | 1.34 | 1 |




## 3.2 HONO observation comparison and pollution patterns

### 3.2.1 HONO observation comparison

For comprehensive comparative analysis, a brief combing of the HONO observations in Beijing was carried out in the past several years, and the results are shown in Table S2 and Figure 3. Overall, the HONO measurement begun to increase since

2016 in Beijing, the time was mainly concentrated in winter (Dec., Jan., and Feb.), and most of which were pollution process analyses. In this work, the observation period was in summer and autumn, and the HONO concentration was comparable to Spataro et al. (2013) and Wang et al. (2017a) (in summer), was obviously lower than Hu et al. (2002), Wu et al. (2009), and Wang et al. (2017a) (in fall), and was obviously higher than Hendrick et al. (2014) and Li et al. (2021). When the air quality was relatively poor in early years, $PM_{2.5}$ and HONO concentrations were both relatively high, now the air quality of Beijing

was improved, both the concentration of $PM_{2.5}$ and HONO decreased. However, during the observation period, $PM_{2.5}$ concentration was low, HONO concentration was still relatively higher compared with other results performed in summer. According to previous studies (Murthy et al., 2020;Du et al., 2013), the narrowing of mixing layer could result in the accumulation of pollutants near the ground, while the increase of mixing layer height (MLH) favored the dilution of pollutants. In general, the MLH was low in winter and high in summer. And the photochemistry was quite active in summer

because of the strong solar radiation, in this case, HONO tended to undergo photolysis. Under the premise that the HONO sources remained constant, the concentration of HONO in summer should be lower than that in winter, and the HONO concentration in autumn should be higher than that in summer. However, our observations showed that HONO concentration in summer was comparable to that in winter and spring, and higher than that in autumn. Wang et al. (2023a) also found that in warm season, the $PM_{2.5}$ and MDA8 $O_3$ increased along with the development of MLH (MLH< 1200 m), they analyzed

that the enhanced secondary chemical formation and increased atmospheric oxidant capacity could explain this phenomenon. Based on this, we discussed the HONO sources in the following sections.



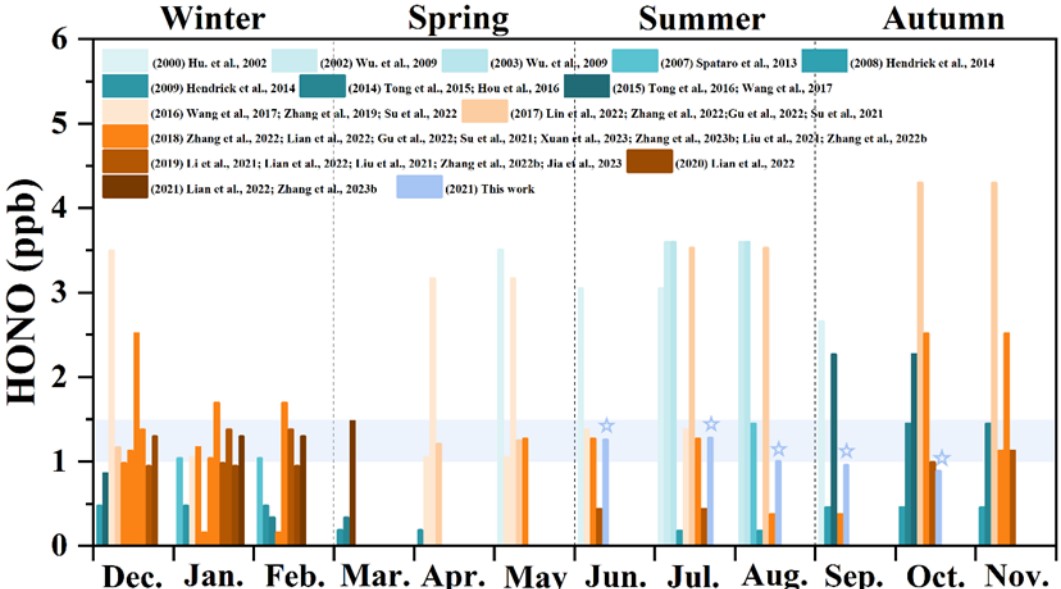

**Figure 3. Overview on HONO measurements performed at Beijing urban sites since 2000. The star symbol represents the averaged HONO concentration measured in this work.**

**3.2.2 HONO variation patterns**

The average diurnal variation of the pollutants during the whole observation period and the averaged values in each month are shown in Figure s4 and Figure 4. It can be seen that the daily averaged pattern of pollutant HONO is that it decreased after sunrise and increased after sunset, this variation pattern was similar to previous studies (Lin et al., 2022;Lian et al., 2022;Zhang et al., 2019a;Zhang et al., 2023b;Zhang et al., 2022). This trend was the same as $NO_2$. NO, $PM_{2.5}$, and CO, that

was, the overall was relatively stable, and it decreased slightly during the daytime. The diurnal variation of $SO_2$ was not large, and the overall concentration was very low, but it had a slightly increase during the daytime. The variation of $O_3$ showed the opposite trend, i.e., its concentration increased after sunrise, peaked in the afternoon and decreased at night. The rapid decrease of HONO in daytime was caused by rapid photolysis and the increase of boundary layer height. The ratio of $HONO/NO_2$ is often used to characterized the heterogeneous reaction of $NO_2$ to form HONO (Yang et al., 2021a;El Zein et

al., 2013;Romanias et al., 2012). At night, the $HONO/NO_2$ ratio was similar to that of HONO, but the ratio increased obviously during the daytime, especially in summer (Jun., Jul., Aug.), and in summer, the light intensity was very strong, as revealed in Figure 4b and 4e. This phenomenon indicated that HONO might have other potential sources related to the solar radiation, e.g., the conversation of $NO_2$ to HONO or nitrate photolysis.

And when HONO was analyzed in months, the characteristics of each month were different. As shown in Figure 4a,

there was a noticeable increase in HONO at noon, most notably in June, followed by September and August, while October had no significant increase. And for the $HONO/NO_2$ ratio in Figure 4b, August, July, June, and September were the months with a significant increase in the noon period, and there was no such phenomenon in October. However, it was important to




note that the NO and $NO_2$ concentration in October were the highest in the entire observation period compared to the low HONO concentration. The NO and $NO_2$ concentrations in June were not high, but the HONO concentration was the highest.

The phenomenon could not be explained by the well-known sources of HONO (Xuan et al., 2023).



**Figure 4.** Diurnal variations in (a) HONO, (b) HONO/NO₂, (c) NO, (d) NO₂, and (e) JNO₂ in each month. The gray shading areas indicate nighttime, 18:00-06:00 LT.





### 3.3 Nocturnal HONO sources and formation

**3.3.1 Direct emission of HONO**

Previous studies showed that the primary emission sources of HONO included biomass burning and vehicles as well as soil emissions (Nie et al., 2015;Su et al., 2011). In this study, soil emission was a negligible source of HONO, since the observation site was located in an urban area, the surrounding soil was not used for agriculture, which greatly reduced HONO emission caused by fertilization process (Su et al., 2011). Except organic compounds (Simoneit, 2002), CO in the gas

phase (Andreae, 2019) and potassium ions ($K^+$) in the aerosol phase (Xinghua et al., 2007) were well recognized inorganic tracers of biomass burning. Many CO sources other than biomass burning, such as industry and traffic, could contribute significantly to the CO loading. Apart from biomass burning emissions, there were no other significant sources of $K^+$ during the measurement. Therefore, $K^+$ was a suitable tracer of biomass burning. The average $K^+$ levels during the five months were 0.14, 0.11, 0.11, 0.15, and 0.13 μg/m$^3$, respectively, lower than 2 μg/m$^3$ (Xu et al., 2019), which indicated that biomass

burning had little effect on this observation site. Hence, only vehicle emission was considered in this study.

The HONO/NOx ratio was used to derive the emission factor in fresh plumes (Kurtenbach et al., 2001). Criteria followed by the fresh plumes were (Yun et al., 2017):

(a) $\triangle NO/\triangle NO_x > 0.9$

(b) good correlation between $NO_x$ and HONO ($R^2 > 0.7$)

(c) short duration of plumes < 1 h

(d) positive correlation between CO and NOx

(e) no precipitation

(f) global radiation < 10 W/m$^2$, or $JNO_2 < 0.25 \times 10^{-3}$ s$^{-1}$

A total of 6 cases met the criteria mentioned above and the details were summarized in Table 2. The mean $\triangle NO/\triangle$

$NO_x$ ratio of the selected fresh plumes was (96.67±4.40) %, which indicated that the chosen air masses were considered to be truly fresh. The linear slope of HONO with $NO_x$ were used as the emission factors of HONO. The correlation coefficients ($R^2$) between HONO and $NO_x$ varied among the cases due to the unavoidably mixing with other air masses, and the range was from 0.72 to 0.93. The obtained $\triangle HONO/\triangle NO_x$ ratios were from 0.32% to 2.49%, with an average value of (1.72± 0.74) %. Thus, a mean $\triangle HONO/\triangle NO_x$ value of 0.017 was used as the emission factor in this work. The $\triangle HONO/\triangle NO_x$

values obtained in urban area of Beijing in previous studies were also summarized in Table S2, the emission factors (EF) reported so far was concentrated in the range of 0.003-0.013, which was slightly lower than the EF in this work. The vehicle engine types, catalytic converters use, and fuels could affect the emission factors of vehicles (Kurtenbach et al., 2001). According to Kessler and Platt (1984), diesel engines had a higher HONO/$NO_x$ ratio than gasoline engines. In our study, the higher HONO/$NO_x$ value possibly due to that the observation site was located outside the fifth ring road, and more heavy-

duty diesel vehicles passed by on the surrounding road at night (regulations of the Public Security Traffic Management



Bureau of the Beijing Municipal Public Security Bureau, i.e., from 6 a.m. to 11 p.m. every day, trucks are prohibited from passing on the roads within the Fifth Ring Road (exclusive), and trucks with an approved load weight of more than 8 tons (inclusive) are prohibited on the main road of the Fifth Ring Road). Assuming that $NO_x$ mainly arose from vehicle emissions, the mean $\triangle HONO/\triangle NO_x$ value of 0.017 was adopted to estimate the vehicle emissions $P_{emis}$ contribution to the HONO
concentration (this factor may lead to an overestimation of the $P_{emis}$ at daytime):

$$P_{emis} = NO_x \times 0.017 \qquad (2)$$

Then the corrected HONO concentration ($HONO_{corr}$) can be obtained from the following equation:

$$HONO_{corr} = HONO - P_{emis} \qquad (3)$$

**Table 2. Emission ratios ($\triangle HONO/\triangle NOx$) of fresh vehicle plumes.**

| Date (yyyy/mm/dd) | Time | $\triangle NO/\triangle NO_x$ | | $\triangle HONO/\triangle NO_x$ | |
|---|---|---|---|---|---|
| | | Slope (%) | $R^2$ | Slope (%) | $R^2$ |
| 2021/6/21 | 2:25-3:15 | 99.72 | 0.999 | 2.492 | 0.7282 |
| 2021/8/31 | 2:20-2:45 | 92.7 | 0.9858 | 2.15 | 0.7625 |
| 2023/9/15 | 3:30-4:20 | 104.11 | 0.9823 | 2.134 | 0.8315 |
| 2023/9/22 | 5:20-5:50 | 95.77 | 0.9717 | 2.018 | 0.8498 |
| 2023/10/3 | 0:40-0:55 | 93.69 | 0.9895 | 1.217 | 0.927 |
| 2023/10/8 | 4:35-5:25 | 94.05 | 0.9929 | 0.323 | 0.7931 |

**3.3.2 Homogeneous formation of HONO**

The homogeneous production of HONO via reaction of NO + OH was also important at night (Li et al., 2012). The net production rate of HONO ($P_{net}$) via homogeneous reaction could be calculated based on the following Equation (4):

$$NO + OH \rightarrow HONO \qquad (R1)$$

$$HONO + OH \rightarrow NO_2 + H_2O \qquad (R2)$$

$$P^{net}_{NO+OH} = k_{NO+OH}[NO][OH] - k_{HONO+OH}[HONO][OH] \qquad (4)$$

where the rate constants of $K_{NO+OH}$ and $K_{HONO+OH}$ for reactions R1 and R2 at 298k were $9.8 \times 10^{-12}$ and $6.0 \times 10^{-12}$ cm$^3$ molecule$^{-1}$ s$^{-1}$, respectively (Liu et al., 2021a). [HONO] and [NO] were the hourly average concentration of HONO and NO at night. The OH concentration during the observation period was estimated according to the previous literature, $2.2 \times 10^5$ molecules cm$^{-3}$ were applied here (Liu et al., 2019;Lin et al., 2022).

The nocturnal variations of HONO, NO, and $P^{net}_{OH+NO}$ during the observation period in each month were shown in Figure 5. Since the values of $K_{HONO+OH}$ and $K_{NO+OH}$ were similar, HONO and NO concentrations were critical to $P^{net}_{OH+NO}$. During the observation period, average values of $P^{net}_{OH+NO}$ were 0.050, 0.045, 0.033, 0.052, and 0.17 ppb/hr, respectively, for June, July, August, September, and October. $P^{net}_{OH+NO}$ had significantly high value in October, which was due to the high NO concentration, as shown in Figure 5a. From 18:00 LT to 22:00 LT in October, $P^{net}_{OH+NO}$ showed an obvious increase, and the
HONO concentration also showed an increase trend, which indicated that homogeneous reaction during this period was the main source of HONO. However, it should be noted that the $P^{net}_{OH+NO}$ in October was the highest, but the concentration of



HONO was particularly lower compared with that in June, July, and September (from 23:00 LT to 6:00 LT), especially that in June, which indicated that the homogeneous reaction in these three month (June, July, and September) was not the main source of HONO.

As shown in Figure 5b, the HONO concentration ranged from 0.96 to 1.47 ppb in the whole observation period. In order to verify the contribution of homogeneous reaction, the accumulated HONO concentration was compared with the integrated $P^{net}_{OH+NO}$ at night in each month. Overall, the contribution of HONO produced by homogeneous reaction to the accumulated HONO concentration during the observation period was 3.38%, 3.47%, 3.44%, 4.58%, and 14.64%, respectively. And this phenomenon suggests that the nocturnal homogeneous process is not the main source of HONO.

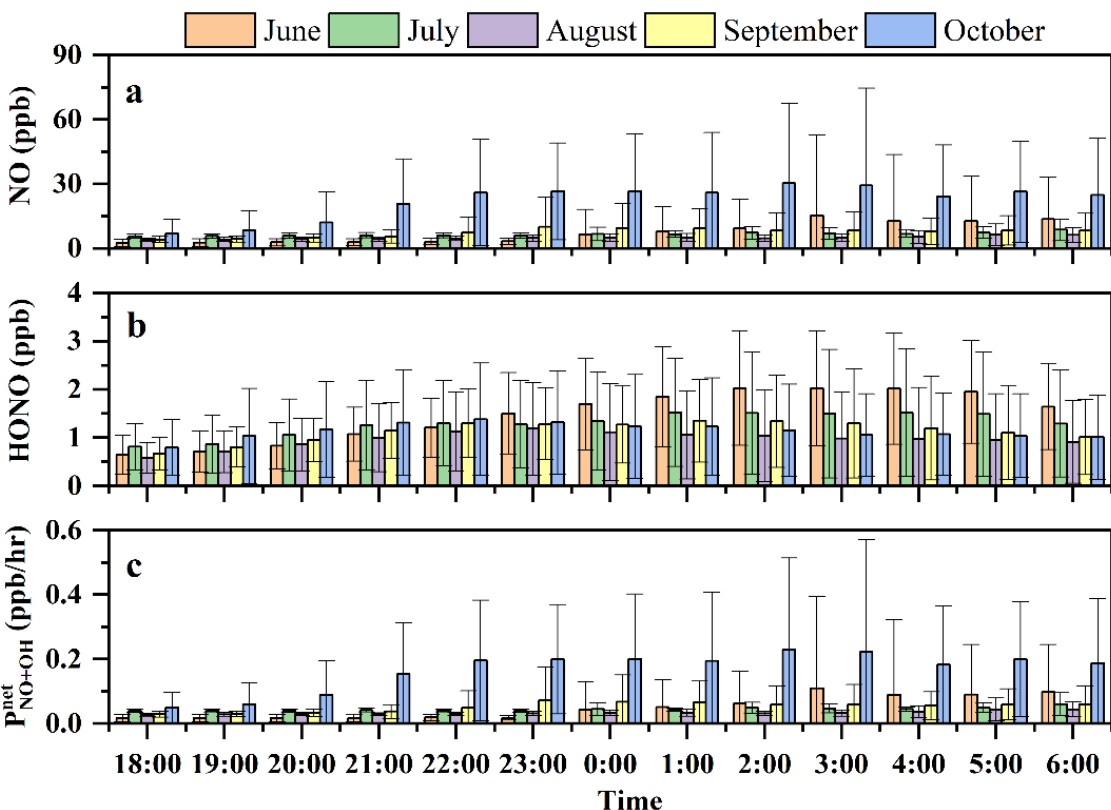

**Figure 5. Homogeneous reaction of OH and NO at night. The error bars represent the standard derivation of NO, HONO, and $P^{net}_{OH+NO}$.**

### 3.3.3 Nighttime heterogeneous conversion of NO₂

Heterogeneous conversion of $NO_2$ on ground or aerosol surface has been recognized as an important HONO source (Finlayson-Pitts et al., 2003;Liu et al., 2020). Nighttime $HONO_{corr}$ concentration could be estimated from the heterogeneous reaction, and the conversion frequency of HONO could be expressed as Equation 7. In order to eliminate the influence of direct emission and diffusion:





$$C_{HONO}^0 = \frac{[HONO_{corr}]_{t2} - [HONO_{corr}]_{t1}}{(t_2 - t_1)\overline{[NO_2]}} \tag{5}$$

$$C_{HONO}^X = \frac{(\frac{[HONO_{corr}]_{t2}}{[X]_{t2}} - \frac{[HONO_{corr}]_{t1}}{X_{t1}})\overline{[X]}}{0.5(t_2-t_1)(\frac{[NO_2]_{t2}}{[X]_{t2}} + \frac{[NO_2]_{t1}}{[X]_{t1}})\overline{[X]}} = \frac{2(\frac{[HONO_{corr}]_{t2}}{[X]_{t2}} - \frac{[HONO_{corr}]_{t1}}{X_{t1}})}{(t_2-t_1)(\frac{[NO_2]_{t2}}{[X]_{t2}} + \frac{[NO_2]_{t1}}{[X]_{t1}})} \tag{6}$$

$$C_{HONO} = \frac{1}{2}(C_{HONO}^0 + C_{HONO}^{CO}) \tag{7}$$

where $\overline{[NO_2]}$ was the mean value of $NO_2$ concentration between time $t_2$ and $t_1$, $C_{HONO}^0$ was the conversion frequency which was not scaled, and $C_{HONO}^X$ was the conversion frequency scaled with reference gases X (CO).

**Table 3. The conversion frequency CHONO, $C_{HONO}[NO_2]$, and $P_{netOH+NO}$ in each month during the observation period.**

|  | June | July | August | September | October |
|---|---|---|---|---|---|
| $C_{HONO}$ (hr$^{-1}$) | 0.011 | 0.0096 | 0.013 | 0.0081 | 0.0017 |
| $C_{HONO}[NO_2]$ (ppb/hr) | 0.15 | 0.089 | 0.082 | 0.11 | 0.033 |
| $P^{net}_{OH+NO}$ (ppb/hr) | 0.034 | 0.035 | 0.034 | 0.046 | 0.15 |

The averaged values of $C_{HONO}$ during the observation period were 0.011 hr$^{-1}$, 0.0096 hr$^{-1}$, 0.013% hr$^{-1}$, 0.0081 hr$^{-1}$, and 0.0017 hr$^{-1}$, respectively, for June, July, August, September, and October, with 0.0087 hr$^{-1}$ on average. The averaged value was slightly higher than that reported by Xuan et al. (2023) and Jia et al. (2020), 0.0073 hr$^{-1}$ and 0.0078 hr$^{-1}$ from August to September, 2018. The varied values of $C_{HONO}$ in each month indicated the different environmental conditions in each month, e.g., surface features, aerosol concentrations (Su et al., 2008). As shown in Table 3, June had the highest $C_{HONO}[NO_2]$, but the lowest $P^{net}_{OH+NO}$, this indicated that the heterogeneous conversion from $NO_2$ was more important for HONO formation at night in June. However, October had the lowest $C_{HONO}[NO_2]$, but the highest $P^{net}_{OH+NO}$, and this indicated HONO produced by homogeneous reaction was more important than heterogeneous conversion from $NO_2$.

In general, the importance of the two generation pathways of nocturnal HONO might change due to the time and environmental factors. The North China has four distinct seasons, temperature and relative humidity varies with the seasons, pollutants in the gas phase and particle phase in the environment can also be affected, and the factors of observation location and observation time should be taken into account when evaluating the relative contribution of the generation pathways to HONO.

### 3.4 HONO daytime budget

### 3.4.1 Budget analysis of HONO

According to the source and sink pathways of HONO mentioned on introduction section, the sources of HONO on the daytime (7:00-18:00 LT) include: (1) direct emission of HONO ($P_{emis}$), (2) homogeneous reaction of NO with OH ($P_{OH+NO}=k_{OH+NO}[OH][NO]$), (3) the unknown sources during the daytime ($P_{unknown}$) (4) the vertical ($T_v$) and horizontal ($T_h$) transport processes of HONO, which were thought to be negligible for the relatively homogeneous atmospheres with calm winds and the intense radiation (Dillon et al., 2002;Sörgel et al., 2011); the sinks of HONO on the daytime include: (1) dry



deposition of HONO ($L_{dep} = \frac{V_{HONO}}{H}[HONO]$), (2) photolysis of HONO ($L_{phot} = J_{HONO}[HONO]$), (3) reaction of OH with

HONO ($L_{OH+HONO} = k_{OH+HONO}[HONO][OH]$).

     The budget of HONO can be calculated by the following equation (8), (9), and (10):

$$\frac{dHONO}{dt} = (P_{emis} + P_{OH+NO} + P_{unknown}) - (L_{dep} + L_{phot} + L_{OH+HONO}) \tag{8}$$

$$P_{unknown} = \frac{dHONO}{dt} + L_{dep} + L_{phot} + L_{OH+HONO} - P_{emis} - P_{OH+NO} \tag{9}$$

$$P_{unkown} = \frac{\Delta HONO}{\Delta t} + k_{OH+HONO}[OH][HONO] + J_{HONO}[HONO]$$


$$+ \frac{V_{HONO}}{H}[HONO] - k_{OH+NO}[OH][NO] - P_{emis} \tag{10}$$

where $\frac{dHONO}{dt}$ represents the difference between HONO sources and sinks; $\frac{\Delta HONO}{\Delta t}$ was the observed change of HONO, the

rate constants of $k_{NO+OH}$ and $k_{HONO+OH}$ at 298k were $9.8 \times 10^{-12}$ and $6.0 \times 10^{-12}$ cm$^3$ molecule$^{-1}$ s$^{-1}$, respectively (Liu et al.,

2021a). A value of $V_{HONO}$ = 1.6 cm/s was adopted here for the deposition rate of HONO (Zhang et al., 2019b), the value of H

= 200 m was used here as the mixing layer height (Hu et al., 2022). The OH concentration and $J_{HONO}$ were calculated with

the method mentioned in section 2.2. According to the model simulation results, the averaged $J_{HONO}$ values were in the range

of $0.78 \times 10^{-3}$ - $1.35 \times 10^{-3}$ s$^{-1}$. The daytime hourly averaged values of OH concentration in the five months were in the ranges

of $2.94 \times 10^6$-$8.88 \times 10^6$ molecules/cm$^3$.

     Figure 6 and S5 illustrated the details production/loss rates and proportion of HONO during the observation period. And

Table 4 showed the averaged production/loss rates of HONO around noontime (10:00-15:00 LT). The dominant loss

pathway of HONO during the five months was the photodecomposition ($L_{phot}$), the averaged values of $L_{phot}$ were 1.10-3.12

ppb/hr around noontime. The following loss pathway of HONO was dry deposition ($L_{dep}$), the averaged values of $L_{dep}$ were

0.13-0.27 ppb/hr around noontime. The loss pathway with the least contribution was the reaction of OH with HONO

($L_{OH+HONO}$), the averaged values of $L_{OH+HONO}$ were 0.029-0.15 ppb/hr at the same time, which was less than 4.8% of that of

$L_{phot}$. For the production pathways of HONO around noontime, the averaged values of homogeneous reaction rate between

NO and OH ($P_{OH+NO}$) were 0.64-1.50 ppb/hr. The averaged values of $P_{emis}$ were 0.16-0.33 ppb/hr. For the high value of $P_{emis}$

in October, the possible reasons maybe that the corresponding diffusion conditions were weakened due to the decrease in

wind speed, and the change of epidemic control policies in this area in October resulted in the increase of the traffic

emissions. The averaged values of $P_{unknown}$ were 0.29-2.37 ppb/hr, the contribution of which to the production of HONO were

74% (June), 52% (July), 61% (August), 62% (September), and 22% (October), respectively. In summary,

photodecomposition was the largest removal pathway around noontime during the whole observation period; $P_{unknown}$

contributed the most to the production of HONO in June, July, August, and September, homogeneous reaction of NO and

OH dominated the HONO production in October at daytime; June had the highest $P_{unknown}$, and October had the lowest

$P_{unknown}$.



The obtained average $P_{unknown}$ values in Summer (from June to August), 1.93 ppb/hr, was at the middle level of those
reported literatures in urban areas: 0.49 ppb/h in Beijing, China from 18 August to 16 September 2018 (Xuan et al., 2023);
0.59 ppb/hr in Beijing, China from June to July 2019 (Li et al., 2021);0.75 ppb/hr in Xi'an, China from 24 July to 6 August
2015 (Huang et al., 2017); 0.98 ppb/h in Nanjing, China from 1 June to 31 August 2018 (Liu et al., 2019); 1.7 ppb/h in
Santiago, Chile from 8-20 March 2005 (Elshorbany et al., 2009); 2.10 ppb/h in Beijing, China from 25 May to 15 July 2018
(Liu et al., 2021a); 2.95 ppb/h in Jinan, China from 1 June to 31 August 2016 (Li et al., 2018); 3.82 ppb/h in Beijing, China
340 from 20 June to 25 July 2016 (Wang et al., 2017a); 4.51 ppb/h in Xiamen, China in August 2018 (Hu et al., 2022). And the
$P_{unknown}$ values in Autumn (from September to October) (0.99 ppb/hr) was at the lower-middle level of those reported
literatures in urban aeras: 0.65 ppb/h in Guangzhou, China from 27 September to 9 November 2018 (Yu et al., 2022b); 2.08
ppb/h in Xiamen, China in October 2018 (Hu et al., 2022); 2.33 ppb/h in Beijing, China from 23 August to 17 September
2018 (Jia et al., 2020); 3.05 ppb/h in Beijing, China from 22 September to 21 October 2015 (Wang et al., 2017a).

As shown in Table 4, the $P_{unknown}$/$NO_2$ ratio and $P_{unknown}$ were not positively correlated, suggesting that there were other
HONO sources besides the heterogeneous reaction of $NO_2$. Particulate nitrate photolysis was one important source of HONO
in the atmosphere (Gen et al., 2022), in June and September, the nitrate concentration was relatively higher than other
months at daytime, as shown in Figure S6, and this could be one important source of HONO in these two months.

**Table 4. The measured $JNO_2$, estimated $JO^1D$, JHONO, OH concentration, averaged production/loss rates and proportion of**
**daytime (10:00-15:00 LT) HONO in each month during the observation period.**

| averaged value | June | July | August | September | October |
|---|---|---|---|---|---|
| $J(NO_2)$ (s$^{-1}$) | $4.02\times10^{-3}$ | $3.42\times10^{-3}$ | $3.78\times10^{-3}$ | $2.99\times10^{-3}$ | $2.83\times10^{-3}$ |
| $J(O^1D)$ (s$^{-1}$) | $1.48\times10^{-5}$ | $1.61\times10^{-5}$ | $1.29\times10^{-5}$ | $1.02\times10^{-5}$ | $0.66\times10^{-5}$ |
| $J(HONO)$ (s$^{-1}$) | $8.19\times10^{-4}$ | $9.07\times10^{-4}$ | $8.08\times10^{-4}$ | $7.96\times10^{-4}$ | $6.72\times10^{-4}$ |
| OH (mole/cm$^3$) | $6.44\times10^6$ | $6.97\times10^6$ | $5.62\times10^6$ | $4.51\times10^6$ | $2.98\times10^6$ |
| $L_{phot}$ (ppb/hr) | 2.81 | 3.12 | 2.16 | 2.45 | 1.10 |
| $L_{dep}$ (ppb/hr) | 0.27 | 0.27 | 0.22 | 0.26 | 0.13 |
| $L_{OH+HONO}$ (ppb/hr) | 0.13 | 0.15 | 0.087 | 0.086 | 0.029 |
| $P_{OH+NO}$ (ppb/hr) | 0.64 | 1.50 | 0.85 | 0.81 | 0.67 |
| $P_{emis}$ (ppb/hr) | 0.18 | 0.19 | 0.16 | 0.21 | 0.33 |
| $P_{unknown}$ (ppb/hr) | 2.37 | 1.84 | 1.57 | 1.69 | 0.29 |
| $P_{unknown}$ /$NO_2$ (hr$^{-1}$) | 0.32 | 0.36 | 0.44 | 0.24 | 0.025 |





**Figure 6. Daytime HONO budget in average production (P$_{emis}$, P$_{NO+OH}$, P$_{unknown}$) and loss rates (L$_{OH+HONO}$, L$_{phot}$, L$_{dep}$) during the five months.**





### 3.4.2 Possible unknown source at daytime

Based on the assumption that nighttime heterogeneous conversion of $NO_2$ continued in the same way at the daytime, the nighttime heterogeneous production of HONO was adopted here, and $P_{het} = C_{HONO}[NO_2]$ (Alicke, 2002;Sörgel et al., 2011). Dark heterogeneous formation accounted for 8.4% (June), 5.71% (July), 6.12% (August), 8.62% (September), and 17.32% (October) of missing sources of HONO at daytime, respectively, which was almost negligible at daytime. And the average value of $P_{unknown}$ normalized by $NO_2$ at daytime (10:00-15:00 LT) was 0.025-0.44 $h^{-1}$, as shown in Table 4, over 15-37 times

greater than the conversion rates at nighttime. This phenomenon implied that $P_{unknown}$ could not be explained by the nocturnal mechanism of $NO_2$ to HONO.

According to the reported possible HONO sources in literatures (Zhang et al., 2023c;Zhang et al., 2022;Lian et al., 2022), light-induced heterogeneous conversion from $NO_2$ to HONO and photolysis of nitrate were responsible for the $P_{unknown}$, solar illumination and relative humidity were the two most frequently considered meteorological factors. Thus, a

correlation analysis was performed to explore the potential unknown sources of HONO at daytime, and the details are shown in Table S3 and Figure S7.

Wang et al. (2017a) analyzed the data of Beijing from 20 June to 25 July 2016 (summer) and found that, compared to $JNO_2$ or RH alone, the correlation between $P_{unknown}$ and product of $JNO_2$ and RH was good. Based on this phenomenon, it was thought that photolytic and heterogeneous reaction occurring upon wet surface were important unknown sources of

HONO. In our work, good correlation between $P_{unknown}$ and product of $JNO_2$ and RH was also found in June, as shown in Table S3, June had the lowest RH and the highest $JNO_2$ value, this phenomenon implied that these combined meteorological factors were conductive to the formation of HONO. However, this phenomenon was not evident in other four months, as they had relative higher RH (due to the precipitation) and lower $JNO_2$ value. Due to the high Henry coefficient of HONO, high humidity conditions might cause the HONO in gas phase to enter the liquid or particle phase, which leaded to a

decrease in the concentration of HONO in the gas phase.

In August and September, it could be seen that low $NO_2$ and $PM_{2.5}$ conditions could lead to high $P_{unknown}$, corresponding to high $JNO_2$ value and relatively low RH (20-40%), as shown in Figure S7. This phenomenon suggested that there were other possible unknown sources of HONO, and this might be due to the light-enhancing effect of $NO_2$ on non-aerosol surfaces.

Light-induced heterogeneous conversion from $NO_2$ to HONO was responsible for the $P_{unknown}$ (Yu et al., 2022a;Jin et al., 2022;Yang et al., 2021a). Thus, the correlation between $P_{unknown}$ and $NO_2$ in the presence of light was analyzed, $JNO_2 \times NO_2 \times PM_{2.5}$ (R = 0.62) > $NO_2 \times PM_{2.5}$ (R = 0.57), $JNO_2 \times NO_2 \times OC$ (R = 0.58) > $NO_2 \times OC$ (R = 0.57), and $JNO_2 \times NO_2 \times EC$ (R = 0.36) > $NO_2 \times EC$ (R = 0.22), in June, this confirms the light-induced heterogeneous enhancement effect.

Wang et al. (2023b) and Ye et al. (2017) found that photolysis of particulate nitrate ($NO_3^-$) was a source of HONO and

$NO_2$ in the troposphere. And the photolysis of nitrate was affected by RH, coexistence of inorganic components, and coexistence of organic components (Gen et al., 2022;Cao et al., 2022). As shown in Table S3, we found significant



correlation between $P_{unknown}$ and $JNO_2 \times [NO_3^-]$ in June (R = 0.48) and July (R = 0.29). Gen et al. (2019) and Zhang et al. (2020) found that the photolysis rate of nitrate particles under high relative humidity conditions could reach an order of $10^{-5}$. In our work, we also found that the correlations between $P_{unknown}$ and $JNO_2 \times [NO_3^-] \times RH$ in June (R = 0.50) and July (R = 0.30) were higher than the conditions without RH.

Through heterogeneous photochemical simulation experiments, Wingen et al. (2008) found that an enhanced $NO_2$ yield was observed as the chloride to nitrate ratio increased. Jin et al. (2022) performed the nitrate photolysis experiments with a flow tube reactor, and found that surprising yields of HONO and NO were formed during the photolysis experiment of $NH_4NO_3$ in the presence of halogen ions ($Cl^-$, $Br^-$, $I^-$), this phenomenon indicated the important role of halogen ions in nitrate photolysis process. Thus, we analyzed the correlation between nitrate and $P_{unknown}$ in the presence of $Cl^-$ ion. As shown in Table S3, the correlation between $P_{unknown}$ and the corresponding factors was as follows: $JNO_2 \times [NO_3^-] \times Cl^-$ (R = 0.46) > $JNO_2 \times [NO_3^-]$ (R = 0.29) in July; $JNO_2 \times [NO_3^-] \times RH \times Cl^-$ (R = 0.44) > $JNO_2 \times [NO_3^-] \times RH$ (R = 0.30) in July; $JNO_2 \times [NO_3^-] \times Cl^-$ (R = 0.093) > $JNO_2 \times [NO_3^-]$ (R = 0.032) in October; $JNO_2 \times [NO_3^-] \times RH \times Cl^-$ (R = 0.075) > $JNO_2 \times [NO_3^-] \times RH$ (R = 0.006) in October. Our observation data also illustrated the facilitating effect of halogen ions on nitrate photolysis.

Bao et al. (2018) found that a large amount of HONO and $NO_x$ were formed during the photolysis experiment of nitrate in the presence of sulfate. Significant positive correlations also occurred in our observations in July and October: $JNO_2 \times [NO_3^-] \times SO_4^{2-}$ (R = 0.32) > $JNO_2 \times [NO_3^-]$ (R = 0.29) in July; $JNO_2 \times [NO_3^-] \times RH \times SO_4^{2-}$ (R = 0.33) > $JNO_2 \times [NO_3^-] \times RH$ (R = 0.30) in July; $JNO_2 \times [NO_3^-] \times SO_4^{2-}$ (R = 0.13) > $JNO_2 \times [NO_3^-]$ (R = 0.032) in October; $JNO_2 \times [NO_3^-] \times RH \times SO_4^{2-}$ (R = 0.16) > $JNO_2 \times [NO_3^-] \times RH$ (R = 0.006) in October. And the enhancement effect in October was significantly stronger than in July.

Yang et al. (2018) and Ye et al. (2019) found that the presence of organic components significantly enhanced the denitrification rate of nitrate. In this work, organic carbon (OC) was used as a proxy for organic components, and the correlation was analyzed. As shown in Table S3, the enhancement of the correlation between $P_{unknown}$ and $NO_3^-$ in the presence of OC was observed, $JNO_2 \times [NO_3^-] \times OC$ (R = 0.55) > $JNO_2 \times [NO_3^-]$ (R = 0.48) in June; $JNO_2 \times [NO_3^-] \times OC$ (R = 0.045) > $JNO_2 \times [NO_3^-]$ (R = 0.032) in October.

Overall, based on the analyzed above, the relatively low humidity and strong solar illumination in June were conductive to the formation of HONO, probably due to the light-induced heterogeneous reaction of $NO_2$. Due to the increase of precipitation in July, the heterogeneous transformation under high humidity conditions was conductive to the formation of HONO, but due to the high Henry coefficient of HONO, more uptake happened for HONO in the air, resulting in the decrease of HONO concentration in the gas phase. The photolysis of nitrate in June and July was also an important source of HONO, probably due to the relatively high humidity and stronger light intensity. In August and September, light-induced formation reactions on non-aerosol surfaces under strong light and low humidity conditions could be one important HONO source. The results of July and October analysis showed that the presence of Cl ions and sulfate significantly enhanced the photolysis of nitrate, resulting in the formation of HONO, and the effect was more obvious in October. The presence of



organic compounds enhanced the photolysis of nitrate, which was more pronounced in June and October. And not only the concentration of HONO but also the source of HONO had temporal patterns, even within the same season, the situation was different from month to month, and this should be considered in model studies.

### 3.4.3 OH production rate at daytime

As HONO was the efficient source of OH radical at daytime (Zhang et al., 2023c;Yu et al., 2022b), the OH production rate from HONO ($P_{OH-HONO}$) was calculated in this work. Ozone photolysis ($P_{OH-O3}$) was also one production pathway of OH radical, and this was calculated and compared to $P_{OH-HONO}$.

$$P_{OH-HONO} = J(HONO)[HONO] \tag{11}$$

$$P_{OH-O3} = 2J(O^1D)[O_3]\Phi_{OH} \tag{12}$$

$$\Phi_{OH} = \frac{k_{O^1D+H_2O}[H_2O]}{k_{O^1D+H_2O}[H_2O]+k_{O^1D+O_2}[O_2]+k_{O^1D+N_2}[N_2]} \tag{13}$$

$$[H_2O] = RH \times \frac{P_{H_2O}}{P} \times N_{air} \tag{14}$$

$$P_{H_2O} = 1013.25 \times EXP\left[13.3185 \times \left(1-\frac{375.15}{273.15+T}\right) - 1.97 \times \left(1-\frac{375.15}{273.15+T}\right)^2 - 0.6445 \times \left(1-\frac{375.15}{273.15+T}\right)^3 - 0.1299 \times \left(1-\frac{375.15}{273.15+T}\right)^4\right] \tag{15}$$

$$[O_2] = 0.20946 \times N_{air} \tag{16}$$

$$[N_2] = 0.78084 \times N_{air} \tag{17}$$

$$N_{air} = \frac{P \times N_A}{R \times (273.15+T)} \tag{18}$$

where [] were the concentration of the substances, $\Phi$ was the OH radical yield in the $O^1D$ reaction pathway, $k_{O^1D+H_2O}$ was $2.2 \times 10^{-10}$ cm$^3$ mole$^{-1}$ s$^{-1}$, $k_{O^1D+O_2}$ was $4.0 \times 10^{-11}$ cm$^3$ mole$^{-1}$ s$^{-1}$, $k_{O^1D+N_2}$ was $2.6 \times 10^{-11}$ cm$^3$ mole$^{-1}$ s$^{-1}$(Atkinson et al., 1997), P was the atmospheric pressure, $P_{H_2O}$ was the partial pressure of water vapor (Seinfeld and Pandis, 2006), $N_{air}$ was the number of molecules of air per unit volume, $N_A$ was $6.022 \times 10^{23}$ mole$^{-1}$ mol$^{-1}$, and R was 8.314 J mol$^{-1}$ K$^{-1}$.

Figure 7 showed that the OH production rate from HONO usually occurred before noontime, and this was caused by the existing HONO and its photolysis, this phenomenon was obvious in July, September, and October. And for $P_{OH-O3}$, the highest value was around noontime, which coincided with $J(O^1D)$ trend, this phenomenon was obvious in June, July, and August. In September and October (autumn), the highest value of $P_{OH-O3}$ occurred a little later after noontime. The average $P_{OH-HONO}$ during 8:00-16:00 LT was 2.70 ppb/h (June), 2.91 ppb/h (July), 2.00 ppb/h (August), 2.25 ppb/h (September), and 0.93 ppb/h (October), respectively; And the average $P_{OH-O3}$ during 8:00-16:00 LT was 0.45 ppb/h (June), 0.82 ppb/h (July), 0.51 ppb/h (August), 0.33 ppb/h (September), and 0.048 ppb/h (October), respectively; the contribution of HONO to OH was significantly greater than that of ozone. The important role of HONO to OH in the atmospheric oxidizing capacity should benefit the production of photochemical ozone (Xuan et al., 2023;Jia et al., 2023;Zhang et al., 2023b;Zhang et al., 2022), the



formation of new particles (Stolzenburg et al., 2023), and the formation of secondary aerosols (Zhang et al., 2023c;Xuan et
al., 2023;Liu et al., 2021a).

Based on the field observations in summer and winter, Liu et al. (2021b) evaluated the atmospheric oxidizing capacity in the megacity of Beijing, and found that the atmospheric oxidizing capacity showed a clear seasonal pattern, which was stronger in summer than that in winter. The dominant oxidant contributor to atmospheric oxidizing capacity at daytime was OH radical, and ozone was the second most important oxidant. Our work showed that in summer and autumn, the
contribution of HONO to OH radical was significantly greater than that of ozone, this further illustrated the importance of HONO. As mentioned in Section 3.4.1, the contributions of $P_{unknown}$ to the production of HONO were 74% (June), 52% (July), 61% (August), 62% (September), and 22% (October), respectively. According to the OH production from HONO, the OH production rate from the missing HONO was also very important to atmospheric oxidation capacity. Thus, further investigation was required to figure out the source of HONO.



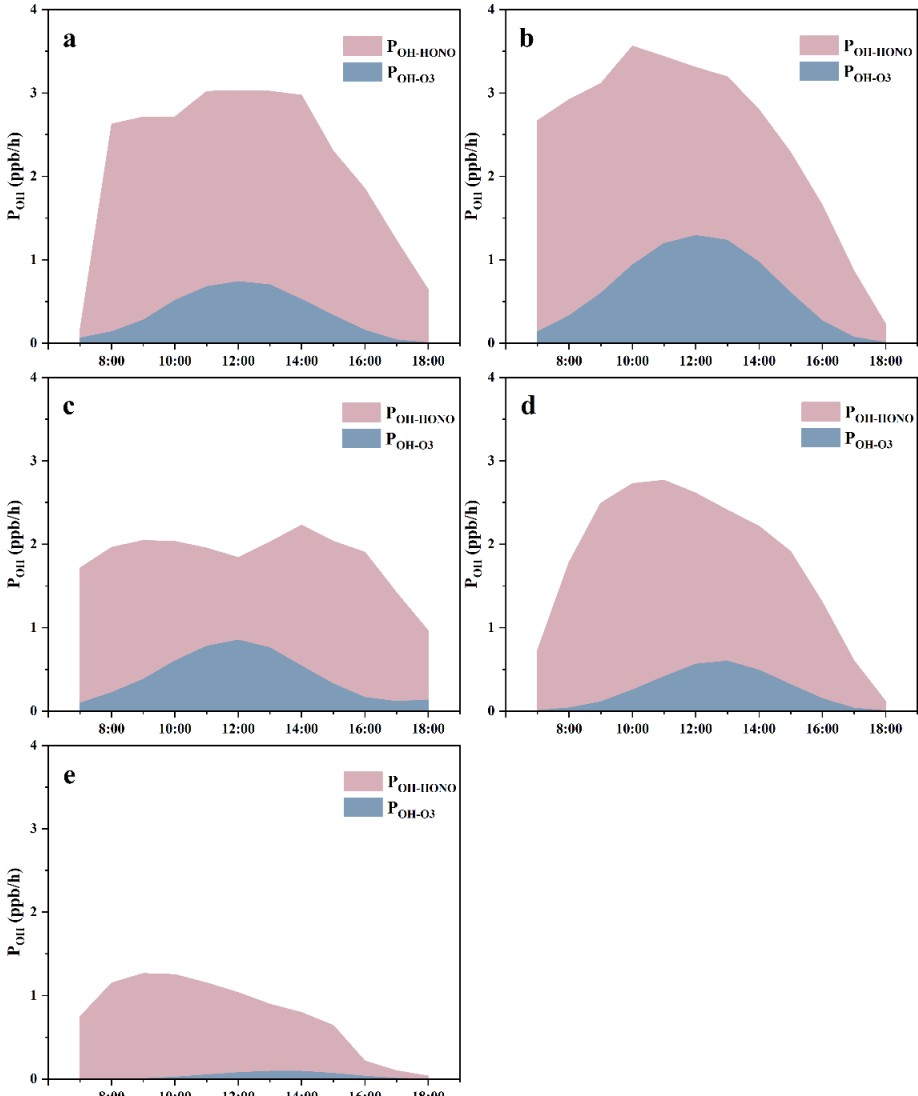


**Figure 7. Averaged OH production rates from photolysis of HONO and ozone in (a) June, (b) July, (c) August, (d) September, and (d) October.**

## 3.5 The correlations between HONO, PM2.5, and O3

In view of the importance of HONO to atmospheric oxidizing capacity (Zhong et al., 2023;Zhang et al., 2023b;Song et al.,

2023), the correlations between HONO, PM2.5, and O3 were analysed. According to the Technical Regulation on Ambient Air Quality Index (on trial) in China (633—2012, 2012), PM2.5 concentrations were divided into three zones: ≤ 35 μg/m³, 35-75 μg/m³, and ≥ 75 μg/m³; and analysed at nighttime and daytime, respectively.

   As shown in Figure 8, PM2.5 concentrations were positively correlated with HONO, i.e., the increase of PM2.5 pollution was accompanied by the increase of HONO concentration during the daytime. At night, except in September, the same





pattern was presented. This phenomenon indicated that the increase in particle pollution in summer and autumn might lead to the formation of HONO and an increase in its concentration. And this was consistent with the sources of HONO analysed in the above sections. However, when it came to ozone, we could see a different phenomenon. In June and July, ozone showed a decreasing trend as particle and HONO concentrations increased during the daytime, but when at night, ozone concentrations tend to increase. In August and September, ozone showed an increasing trend as particle and HONO

concentration increased both during the daytime and nighttime. In October, ozone showed an increasing trend at daytime, but a decreasing trend at night. Wang et al. (2017b) summarized the ozone abundance and its relationship to chemical processes and atmospheric dynamics in China, the ozone formation and concentration could be affected by various factors, including precursors, meteorological factors, and regional transport, et al., the formation of OH radical by HONO photolysis was a pathway involved in the $RO_x$ and $NO_x$ cycle of ozone formation, ozone was not directly related to HONO.

Wang et al. (2023a) analysed the relationships between ozone, $PM_{2.5}$, and mixing layer height in warm season, and found that the enhanced atmospheric oxidant capacity could promote the secondary transformation of particles, and weakened the dilution effect of mixing layer height rise on pollutants. The contribution of observed HONO to atmospheric oxidant capacity exceeded that of ozone, so it was necessary to pay more attention to the sources of HONO in warm season.

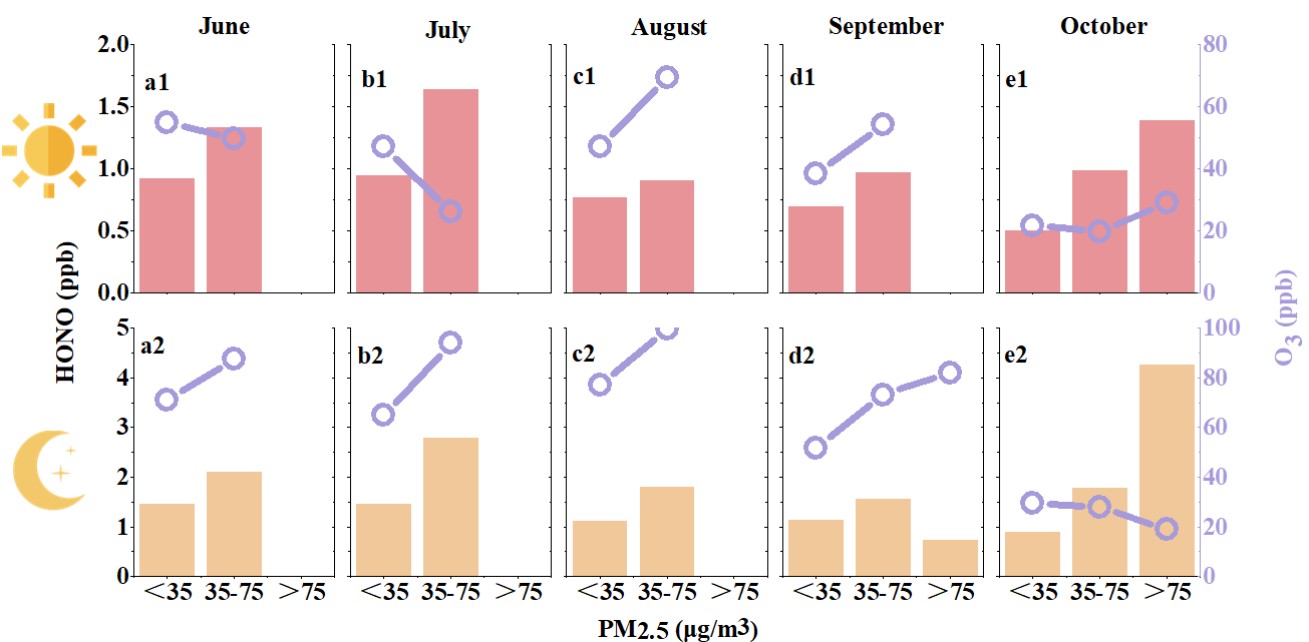

**Figure 8. Distributions of HONO and O₃ mean concentrations under different PM2.5 pollution conditions. The upper panel was the daytime average value (7:00-18:00 LT), and the bottom panel was the nighttime average value (19:00-6:00 LT).**



## 4 Conclusions

Continuous field observation of HONO in warm seasons was conducted in an urban site of Beijing, from June to October 2021. The monthly average HONO concentration was in the range of 0.89-1.28 ppb, showing a larger contribution to OH radical relative to ozone at daytime, above three times as that of ozone in each month. Compared with previous field observations in the urban sites of Beijing, the contribution of vehicle emissions to HONO (0.017) was relatively high. The monthly nocturnal conversion rate of $NO_2$ to HONO was in the range of 0.0017-0.093 $hr^{-1}$, accounted for 5.71%-17.32% of missing sources of HONO at daytime; the average value of $P_{unknown}$ normalized by $NO_2$ at daytime (10:00-15:00 LT) was 0.025-0.44 $hr^{-1}$, over 15-37 times greater than the conversion rates at nighttime, which indicated that nocturnal heterogeneous conversion from $NO_2$ to HONO was almost negligible at daytime. Light-induced heterogeneous conversion of $NO_2$ to HONO under medium and low RH condition and photolysis of nitrate were responsible for $P_{unknown}$ at daytime, RH and solar illumination were two important influencing factors. In addition, differences in the components of particles, e.g., halogen ions, sulfate, organic aerosol, could also lead to changes in the concentration of HONO formation.

In recent years, the concentration of atmospheric particulate matter in China decreased significantly, but the ozone concentration showed a fluctuating upward trend, the atmospheric oxidation capacity increased significantly, especially in the warm season. Given the contribution of HONO to atmospheric oxidation capacity, its sources should be studied in more detail.

**Code and data availability.** The data used in this study are available upon request from the corresponding author.

**Author contributions.** JLL and HL conceived and led the studies. JLL, CFL, HZ, WGW, and HL performed observation studies and data analysis. MML, YXY, YFS, CC, YCG, YSX, JG, and MFG discussed the results and commented on the paper. JLL prepared the article with contributions from all co-authors.

**Competing interests.** The authors declare that they have no conflict of interest.

**Financial support.** This research has been supported by National Key Research and Development Program of China (No. 2023YFC3706102), National Natural Science Foundation of China (NSFC, No. 41931287, No.42130606, No. 42175133, No. 42307139).

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
