# Peer review of "Characterization of nitrous acid and its potential effects on secondary pollution in warm-season of Beijing urban areas"

_EGUsphere, 2024_

## Referee Comment (RC1)

Review of Li, et al., ACP-2024-367

This paper presents HONO and associated measurements from a ground site in Beijing during the months of June to October, 2021. There is an attempt to assess non-traditional sources of HONO (sources aside from OH + NO) using co-measured NOx and particle loading. There is a serious flaw in the paper that cannot be overcome by revision or further analysis (see below). The full paper as submitted here should be rejected for publication. However, the authors should consider publishing the HONO data and associated summary of measurements from Beijing (there have been many) as an ACP Measurement Report. I have the following General and Specific comments.

General comments.

The major flaw in this paper is there is only an NOy measurement at the site, which we know to measure not just NOx but also PAN, $HNO_3$, particle, and alkyl nitrates (NOz compounds). The authors try to argue that the impact of NOz compounds is minor. We know that this is not true, especially for the mid-day period when $NO_2$ is below 10 ppbv, and there is obvious $O_3$ production (see for example Zhang et al., 2015, Zhang et al., 2023 (which shares some authors with this paper). Under these conditions in particular, equating $NO_2$ with NOz will result is errors of factors of 2 -3 at least. All of the interpretation that the authors try to do with this data is fatally flawed.

We are given essentially no details about the measurement site and are given only references to describe the HONO measurement. So, we have no idea if the method has interferences, from other N compounds aside from $NO_2$. We have no idea what materials that might from or store HONO (soil, asphalt) surround the site.  At least a brief description of these is necessary.

The authors basically prescribe nighttime OH, and broadly parameterize daytime OH. We know that there is substantial variability in OH, so these shortcuts will mask much of the chemical dependencies that the authors are trying to uncover in their analyses.

In several places the authors try to use slight differences in $R^2$ to say something about what factors might be more responsible for HONO. In some cases these $R^2$ values are below 0.1, which means neither factor was significant. Such an analysis doesn't tell us anything.

Specific comments

The abstract is too long and has too much detail.
Line 20. What are the units of $P_{emis}$?
Line 29. The solubility of HONO is very much pH dependent. Below its pKa (3.28) it is not very soluble.
Line 55. The authors have missed VandenBoer, et al., 2015.
Line 70. This sentence is meaningless.
Line 89. The authors did not tell us what the impact of their measured HONO on $O_3$ production was.
Line 113. "was" should be "were"

Line 119. OH is going to vary a lot, with dependencies that are very much at the heart of what the authors are trying to get at. This specification is not very useful.

Line 141-142. What were the averaging times for these?

Line 158. Don't know what a "brief combing" is.

Line 194-195. This isn't what Figure 4a shows.

Line 216. Emission factor of what?

Line 225. Too many significant figures, only 3 at most are justified.

Line 241. What are the units?

Line 268. Too many significant figures, only 2 (3 for the last number) at most are justified.

Line 280. There is no explanation and justification for this equation.

Line 294. You haven't described the observation location.

Line 3456. The averaging here (monthly) is so broad as to make this statement meaningless.

Line 373. See previous comment about Henry's coeff.

Line 425-435. This could all go in the SI, there is nothing new here.

Line 450-459. This could all go in the Conclusions.

References

VandenBoer, et al., Nat. Geosci., 8, 55-60, 2015.

Zhang, G., et al., Atmos Environ., 103, 289-296, 2015.

Zhang, H., et al., Sci. Total Environ., 905, 166852, 2023.

---

## Author Comment (AC1)

**Response to the comments of Reviewer #1**

*This paper presents HONO and associated measurements from a ground site in Beijing during the months of June to October, 2021. There is an attempt to assess non-traditional sources of HONO (sources aside from OH + NO) using co-measured NOx and particle loading. There is a serious flaw in the paper that cannot be overcome by revision or further analysis (see below). The full paper as submitted here should be rejected for publication. However, the authors should consider publishing the HONO data and associated summary of measurements from Beijing (there have been many) as an ACP Measurement Report. I have the following General and Specific comments.*

Response: We thank Referee #1 for the review and the evaluation of our manuscript. We have fully considered the comments and responded to these comments below in blue text. The revisions in the manuscript are highlighted in yellow color. The response and changes are listed below.

*General comments.*

1. *The major flaw in this paper is there is only an $NO_y$ measurement at the site, which we know to measure not just $NO_x$ but also PAN, $HNO_3$, particle, and alkyl nitrates (NOz compounds). The authors try to argue that the impact of NOz compounds is minor. We know that this is not true, especially for the mid-day period when $NO_2$ is below 10 ppbv, and there is obvious $O_3$ production (see for example Zhang et al., 2015, Zhang et al., 2023 (which shares some authors with this paper). Under these conditions in particular, equating $NO_2$ with $NO_z$ will result is errors of factors of 2 -3 at least. All of the interpretation that the authors try to do with this data is fatally flawed.*

   Actually, due to the employment of a molybdenum $NO_2$-to-NO converter, the 42i analyzer might overestimate $NO_2$ concentration for the potential convertion of NOz (NOz = NOy – NOx. e.g., HONO, $HNO_3$, peroxyacetyl nitrate (PAN), and so on). Based on the articles mentioned (Zhang et al., 2023; Zhang et al., 2015) above, we conducted a period of targeted observations, and two devices were applied: the Teledyne API Model N500 CAPS NOx analyzer and the Thermo Scientific 42i analyzer; the Teledyne API Model N500 CAPS NOx analyzer is a Cavity Attenuated Phase Shift (CAPS), non-chemiluminescent NOx analyzer, which can directly measure $NO_2$.

| the instruments applied in this targeted observation |
|---|

[Figure]

| N500 NO$_X$-NO$_2$-NO analyzer | 42i NO-NO$_2$-NO$_X$ analyzer |
|---|---|

The observation period was from May 19th to 31st, 2024. The specific weather conditions these days are shown in the figure below.

| Sun. | Mon. | Tues. | Wed. | Thurs. | Fri. | Sat. |
|---|---|---|---|---|---|---|
| 19 | 20 | 21 | 22 | 23 | 24 | 25 |
| 26 | 27 | 28 | 29 | 30 | 31 | 1 |

Note: green-excellent, yellow-good, orange-lightly polluted (ozone)

The comparison plot of NO$_2$ data during the observation period is shown in the figures below. When the NO$_2$ concentration is above 7 ppb, there is no significant difference between the two instruments. However, when NO$_2$ concentration is below 7 ppb, the 42i values are slightly higher than those of the N500, with a corresponding slope of 1.14. Based on this measurement, we have corrected all NO$_2$ values below 7 ppb during the observation period, as well as the corresponding analysis. We have also provided an explanation in the methods section. (Page 4, line 96-102; SI, Page 7, Figure S1)

[Figure]

References:

Zhang, G., Mu, Y., Zhou, L., Zhang, C., Zhang, Y., Liu, J., Fang, S., and Yao, B.: Summertime distributions of peroxyacetyl nitrate (PAN) and peroxypropionyl nitrate (PPN) in Beijing: Understanding the sources and major sink of PAN, Atmospheric Environment, 103, 289-296, 10.1016/j.atmosenv.2014.12.035, 2015.

Zhang, H., Tong, S., Zhang, W., Xu, Y., Zhai, M., Guo, Y., Li, X., Wang, L., Tang, G., Liu, Z., Hu, B., Liu, C., Liu, P., Sun, X., Mu, Y., and Ge, M.: A comprehensive observation on the pollution characteristics of peroxyacetyl nitrate (PAN) in Beijing, China, Science of The Total Environment, 905, 166852, 10.1016/j.scitotenv.2023.166852, 2023.

2. *We are given essentially no details about the measurement site and are given only references to describe the HONO measurement. So, we have no idea if the method has interferences, from other N compounds aside from $NO_2$. We have no idea what materials that might from or store HONO (soil, asphalt) surround the site. At least a brief description of these is necessary.*

We thank the reviewer for pointing this out. The description of this site has been added in the methods part of the main text.

(Page 3-4, line 87-92) "Briefly, Chaoyang District was located at the eastern area of Beijing and was one of the six main urban districts (Dongcheng, Xicheng, Haidian, Chaoyang, Shijingshan, and Fengtai) of Beijing. This site was located about 2 km to the north of the North Fifth Road, approximately 0.3 km away from Beiyuan Road, with a high volume of traffic. And this site was located in a mixed-use commercial and residential area, with several shopping malls, residential areas, and office buildings nearby, and there were no obvious sources of industrial pollution. Thus, this site could be considered as an urban site."

3. *The authors basically prescribe nighttime OH, and broadly parameterize daytime OH. We know that there is substantial variability in OH, so these shortcuts will mask much of the chemical dependencies that the authors are trying to uncover in their analyses. In several places the authors try to use slight differences in $R^2$ to say something about what factors might be more responsible for HONO. In some case these $R^2$ values are below 0.1, which means neither factor was significant. Such an analysis doesn't tell us anything.*

Actually, OH concentrations in the atmosphere have high variability. The empirical model applied to calculate the daytime OH concentration was proposed by Rohrer and Berresheim (2006), they found the strong nearly linear correlations between measured OH concentrations and simultaneously observed $J(O^1D)$, despite the fact that OH concentrations were influenced by thousands of reactants. Liu et al. (2019) analyzed this calculation method, the calculated OH concentrations around noon were comparable to the observations in Chinese urban or suburban atmospheres. Based on the research background above, we applied this calculation formula to provide a visual assessment of OH concentrations during the observation period. We have added explanations accordingly in the manuscript. Indeed, a very small R-squared ($R^2$) value did not explain much, the contents with analysis that $R^2 < 0.1$ were removed.

[revised manuscript text omitted]

6. *Line 29. The solubility of HONO is very much pH dependent. Below its pKa (3.28) it is not very soluble.*

   Thank you for this comment. We have revised the interpretation in the manuscript.

7. *Line 55. The authors have missed VandenBoer, et al., 2015.*

Thank you for this comment. This has been added in the manuscript.

(Page 2, line 46-47) "The release from soil nitrite is one important primary source of HONO (Su et al., 2011; VandenBoer et al., 2015; Wu et al., 2019)"

(Page 26, Line 667-669) *"VandenBoer, T. C., Young, C. J., Talukdar, R. K., Markovic, M. Z., Brown, S. S., Roberts, J. M., and Murphy, J. G.: Nocturnal loss and daytime source of nitrous acid through reactive uptake and displacement, Nat. Geosci., 8, 55-60, 10.1038/ngeo2298, 2015."*

8. *Line 70. This sentence is meaningless.*

This sentence has been removed.

9. *Line 89. The authors did not tell us what the impact of their measured HONO on $O_3$ production was.*

The correlation between HONO and $O_3$ was analyzed in Section 3.5. We found that the correlation between ozone and HONO varies in different months, corresponding explanations was added in the manuscript.

(Page 21, line 443-445) "……the ozone formation and concentration could be affected by various factors, including precursors, meteorological factors, and regional transport, et al., the formation of OH radical by HONO photolysis was a pathway involved in the ROx and NOx cycle of ozone formation, ozone was not directly related to HONO."

10. *Line 113. "was" should be "were"*

This has been corrected. (Page 4, line 111)

11. *Line 119. OH is going to vary a lot, with dependencies that are very much at the heart of what the authors are trying to get at. This specification is not very useful.*

Actually, OH concentrations in the atmosphere have high variability. The empirical model proposed by Rohrer and Berresheim (2006) was applied here to calculate the daytime OH concentration. Rohrer and Berresheim (2006) found the strong nearly linear correlations between measured OH concentrations and simultaneously observed $J(O^1D)$, despite the fact that OH concentrations were influenced by thousands of reactants. Liu et al. (2019) analyzed this calculation method, the calculated OH concentrations around noon were comparable to the observations in Chinese urban or suburban atmospheres. Based on the research background above, we applied this calculation formula to provide a visual assessment of OH concentrations during the observation period. And we have added explanations accordingly in the manuscript.

(Page 4-5, line 118-123) "……these coefficients adopted here were from the OH studies in the Pearl River delta (PRD) and Beijing, China (Rohrer et al., 2014; Tan et al., 2017, 2018). According to the summarizing coefficients in different OH observation campaigns in the polluted areas of China (Liu et al., 2019), the comprehensive impact of reactants (e.g., VOCs and NOx) on OH could not compete with that of UV light to OH, the chemical environments of OH could be similar. This could be a reasonable way to derive OH concentration by the equation above."

12. *Line 141-142. What were the averaging times for these?*

    The averaging times referred to here is for the entire observation period, this has been added in the manuscript. (Page 6, Table 1)

13. *Line 158. Don't know what a "brief combing" is.*

    Thank you for this comment. This has been corrected in the manuscript.

    (Page 7, line 161) "For comprehensive comparative analysis, a review of the HONO observations in Beijing was carried out……"

14. *Line 194-195. This isn't what Figure 4a shows.*

    This part of the content has been removed from the manuscript. (Page 8, line 192)

15. *Line 216. Emission factor of what?*

    This has been added in the manuscript.

    (Page 10, line 213) "……the emission factor of vehicles……"

16. *Line 225. Too many significant figures, only 3 at most are justified.*

    Thank you for this comment. All the significant figures in the manuscript have been modified.

17. *Line 241. What are the units?*

    This has been added in the manuscript.

    (Page 11, line 236) "……$\triangle$HONO/$\triangle$NO$_x$ value of 0.017 was adopted to estimate the vehicle emissions $P_{emis}$ (ppb/h) contribution to……"

18. *Line 268. Too many significant figures, only 2 (3 for the last number) at most are justified.*

    Thank you for this comment. We have made the corresponding modifications in the manuscript.

19. *Line 280. There is no explanation and justification for this equation.*

    This has been added in the manuscript.

    (Page 13, line 286-288) "Then the NO$_2$ to HONO conversion rate ($C_{HONO}$) was calculated by the combination of $C_{HONO}^0$ (not scaled $C_{HONO}$) and $C_{HONO}^{CO}$ (CO scaled $C_{HONO}$), which could reduce the

impact of uncertainties in diffusion process and emissions on the conversion rate."

20. *Line 294. You haven't described the observation location.*

This has been added in the manuscript.

(Page 3-4, line 87-92) "Briefly, Chaoyang District was located at the eastern area of Beijing and was one of the six main urban districts (Dongcheng, Xicheng, Haidian, Chaoyang, Shijingshan, and Fengtai) of Beijing. This site was located about 2 km to the north of the North Fifth Road, approximately 0.3 km away from Beiyuan Road, with a high volume of traffic. And this site was located in a mixed-use commercial and residential area, with several shopping malls, residential areas, and office buildings nearby, and there were no obvious sources of industrial pollution. Thus, this site could be considered as an urban site."

21. *Line 3456. The averaging here (monthly) is so broad as to make this statement meaningless.*

The statement here has been removed. And this table has been removed to SI. (SI, table S3)

22. *Line 373. See previous comment about Henry's coeff.*

The corresponding explanations in the manuscript have been revised.

23. *Line 425-435. This could all go in the SI, there is nothing new here.*

This has been removed to the SI. (SI, page 1, line 8-19)

24. *Line 450-459. This could all go in the Conclusions.*

This part of the content has been moved to the Conclusions. (Page 22, line 465-473)

---

## Author Comment (AC2)

*Referee #2*

*The study by Li et al. provides an extensive record of HONO measured during a field campaign Beijing, China in 2021. The novelty of this work comes from the fact that it HONO concentration measurements during the summer and autumn months, which have been lacking from previous studies conducted in Beijing, which have mostly occurred during the winter months (Figure 3). Detection of HONO was conducted using a LOPAP system. Analysis of the data was somewhat routine was focused on evaluating potential nighttime and daytime sources of HONO during the campaign, in addition to determining the impact of HONO relative to other OH sources on the oxidative capacity in the region. This approach is typical of many papers that attempt to determine the relative influence of the various HONO sources on observed ambient concentrations. The conclusions or analysis approaches are not novel. After calculating a rate of HONO formation from the unknown daytime source, there is some speculation that it is from photo-enhanced $NO_2$ conversion or nitrate photochemistry, which may be supported by some of the data, depending on the month. The work is valuable as a record of HONO concentrations from an important urban area during a time of year that is less well studied and alone for that should be probably be published eventually-- after the manuscript is revised for clarity, based on the suggestions below.*

Response: We thank Anonymous Referee #2 for the review and the positive evaluation of our manuscript. We have fully considered the comments and responded to these comments below in blue text. The revisions in the manuscript are highlighted in yellow color. The response and changes are listed below.

1. *Significant figures: There are numerous cases within the text and in tables where too many significant figures are used when reporting numbers (e.g., Table 1 or section 3.1.1, reporting temperature to the hundredth of a degree, or relative humidity to a hundredth of a percent; section 3.1.2, trace gas measurements, etc. many of these measurements are likely not accurate out to that many decimal points and the values should be rounded off appropriately.*

   Thank you for this comment. We have made the corresponding modifications in the manuscript.

2. *Figure 2: This figure was of very poor quality such that it was very difficult to read. The resolution was very low and colors chosen (e.g., yellow or pink) were of low contrast, making it almost impossible to read.*

We have made modifications to Figure 2. (Page 6, line 155)

[Figure]

3. *Section 3.2.1 and Figure 3: I feel it is difficult to make comparisons between HONO concentrations made during different seasons over a 20 year period in Beijing based simply on monthly averages. Error bars or any other indicator of variation in the data is not indicated for these values and it is not clear whether median concentrations may be a better way to report the data. Without consideration of the variation of these concentrations, it is not possible to make conclusions about whether values in summer are higher (in a statistically significant way) than in autumn or winter, etc.*

   To more accurately describe the data reported in the literatures, we have removed figure 3 in the manuscript, the error bars and the other indicators of variation in the data were added in Table S2. The corresponding context in the manuscript has been updated, due to the focus on different aspects in the observations, some on pollution processes and others on longer time scales, we have removed the content related to comparisons from the main text. (SI, Table S2) (main text, page 7, line 161-177)

4. *Line 242: I found the term "corrected HONO concentration (HONO_{corr}) confusing. It would help to explain that this is the concentration of HONO in air that is not due to direct vehicular emissions.*

   We have added the explanation in the manuscript.

   (Page 11, line 239-240) "Then the concentration of HONO in air that is not due to direct vehicular emissions (the corrected HONO concentration, HONO_{corr}) can be obtained from the following equation……"

5. *Line 251: Symbols for the rate constants should be written with lower case "k" instead of capital letter, which would be understood as an equilibrium constant.*

This has been corrected in the manuscript.

(Page 11, line 249) "……where the rate constants of $k_{NO+OH}$ and $k_{HONO+OH}$ for reactions R1 and R2……"

6. *Line 275: $HONO_{corr}$ is here referred to as the HONO concentration due to heterogeneous $NO_2$-to-HONO conversion during the nighttime. However, in equation (3) it is all HONO that is not due to direct vehicular emissions. Perhaps a different symbol or term should be used for referring to the nighttime HONO concentrations due solely to $NO_2$ heterogeneous reaction to avoid confusion.*

This has been modified in the manuscript.

(Page 12, line 273) "Nighttime $HONO_{het,night}$ concentration could be estimated……"

(Page 13, line 282-284)

$$"k_{HONO}^0 = \frac{[HONO_{corr,night}]_{t2} - [HONO_{corr,night}]_{t1}}{(t_2-t_1)[NO_2]} \qquad (5)$$

$$k_{HONO}^X = \frac{(\frac{[HONO_{corr,night}]_{t2}}{[X]_{t2}} - \frac{[HONO_{corr,night}]_{t1}}{X_{t1}})\overline{[X]}}{0.5(t_2-t_1)(\frac{[NO_2]_{t2}}{[X]_{t2}} + \frac{[NO_2]_{t1}}{[X]_{t1}})\overline{[X]}} = \frac{2(\frac{[HONO_{corr,night}]_{t2}}{[X]_{t2}} - \frac{[HONO_{corr,night}]_{t1}}{X_{t1}})}{(t_2-t_1)(\frac{[NO_2]_{t2}}{[X]_{t2}} + \frac{[NO_2]_{t1}}{[X]_{t1}})} \qquad (6)$$

$$k_{HONO,het-night} = \frac{1}{2}(k_{HONO}^0 + k_{HONO}^{CO}) \qquad (7)"$$

7. *Equations 5-7: The rational/derivation of these equations is not clear and symbolism is very unclear and there are several typos in the equations. Besides the [$HONO_{corr}$] term described above, it was confusing to use the symbol "C" for a conversion frequency since C is used often to represent concentration, and the units of the "conversion frequency" suggest they are first-order rate constants. Also, it is not clear why the conversion frequencies are scaled to CO concentrations. A clarification would be useful here.*

Thank you for this suggestion, the content mentioned in the comment has been modified in the manuscript. The explanation of rational/derivation of these equations has been added, and the symbol for a conversion frequency has been corrected to "k", the reasons for the application of CO concentration is also added in the manuscript.

(Page 12-13, line 273-288)

"Nighttime $HONO_{het-night}$ concentration could be estimated from the heterogeneous reaction (R3, the mechanism of heterogeneous formation of HONO, and this was first order in $NO_2$ and $H_2O$ (Alicke

et al., 2002)), and the conversion frequency of HONO ($k_{HONO,het\text{-}night}$) could be expressed as Equation 7. We determined the HONO formation by assuming a linear increase of its mixing ratio during a time interval ($t_2$-$t_1$). Since the mechanism summarized in R3 was first order in $NO_2$, the HONO formation was proportional to the $NO_2$ concentration. The conversion frequency was also assumed to be independent of gas phase water (Kleffmann et al., 1998), the average nighttime conversion frequency was determined by Equation 5,6, and 7. In order to eliminate the influence of direct emission and diffusion, CO was chosen as the reference species used for normalization:

$$2NO_2 + H_2O \xrightarrow{\text{ground,aerosol surface}} HONO + HNO_3 \tag{R3}$$

$$k_{HONO}^0 = \frac{[HONO_{corr,night}]_{t2} - [HONO_{corr,night}]_{t1}}{(t_2 - t_1)\overline{[NO_2]}} \tag{5}$$

$$k_{HONO}^X = \frac{(\frac{[HONO_{corr,night}]_{t2}}{[X]_{t2}} - \frac{[HONO_{corr,night}]_{t1}}{X_{t1}})\overline{[X]}}{0.5(t_2-t_1)(\frac{[NO_2]_{t2}}{[X]_{t2}} + \frac{[NO_2]_{t1}}{[X]_{t1}})\overline{[X]}} = \frac{2(\frac{[HONO_{corr,night}]_{t2}}{[X]_{t2}} - \frac{[HONO_{corr,night}]_{t1}}{X_{t1}})}{(t_2-t_1)(\frac{[NO_2]_{t2}}{[X]_{t2}} + \frac{[NO_2]_{t1}}{[X]_{t1}})} \tag{6}$$

$$k_{HONO,het-night} = \frac{1}{2}(k_{HONO}^0 + k_{HONO}^{CO}) \tag{7}$$

where $\overline{[NO_2]}$ was the mean value of $NO_2$ concentration between time $t_2$ and $t_1$, $k_{HONO}^0$ was the conversion frequency which was not scaled, and $k_{HONO}^X$ was the conversion frequency scaled with reference gases X (CO). Then the $NO_2$ to HONO conversion rate ($k_{HONO}$) was calculated by the combination of $k_{HONO}^0$ (not scaled $k_{HONO}$) and $k_{HONO}^{CO}$ (CO scaled $k_{HONO}$), which could reduce the impact of uncertainties in diffusion process and emissions on the conversion rate."

Reference (Page 24, line 590-591)

Kleffmann, J., Becker, K. H., and Wiesen, P.: Heterogeneous $NO_2$ conversion processes on acid surfaces: Possible atmospheric implications, Atmos. Environ., 32, 2721-2729, 10.1016/s1352-2310(98)00065-x, 1998.

8. *Table 3: This table compares HONO conversion frequencies and production rates and forms the basis of a comparison. I recommend including errors and when comparing values from this study to others, one should conduct and report results of the appropriate statistical tests of significance.*

Thank you for this suggestion. The errors have been added in the manuscript.

9. *Lines 334-343: This paragraph compares the production rate of HONO due to "unknown sources" derived from this work to values previously reported in the literature. It is one continuous string of values with references and as such is extremely difficult to read. I recommend including all this information in a table or figure to facilitate comparison.*

Thank you so much for this suggestion. This paragraph has been modified and all the information has been included in a table.

(Page 15, line 344)

"Table 4. The $P_{unknown}$ values in this work and reported literatures."

| Date | | value (ppb/h) | location | literatures |
|---|---|---|---|---|
| Summer | 18 August to 16 September, 2018 | 0.49 | Beijing | Xuan et al., 2023 |
| | June to July, 2019 | 0.59 | Beijing | Li et al., 2021 |
| | 24 July to 6 August, 2015 | 0.75 | Xi'an | Huang et al., 2017 |
| | 1 June to 31 August, 2018 | 0.98 | Nanjing | Liu et al., 2019 |
| | 8-20 March, 2005 | 1.7 | Santiago | Elshorbany et al., 2009 |
| | 25 May to 15 July, 2018 | 2.1 | Beijing | Liu et al., 2021a |
| | June to August, 2021 | 2.3 | Beijing | This work |
| | 1 June to 31 August, 2016 | 3.0 | Jinan | Li et al., 2018 |
| | 20 June to 25 July, 2016 | 3.8 | Beijing | Wang et al., 2017a |
| | August, 2018 | 4.5 | Xiamen | Hu et al., 2022 |
| Autumn | 27 September to 9 November, 2018 | 0.65 | Guangzhou | Yu et al., 2022b |
| | September to October, 2021 | 1.0 | Beijing | This work |
| | October, 2018 | 2.1 | Xiamen | Hu et al., 2022 |
| | 23 August to 17 September, 2018 | 2.3 | Beijing | Jia et al., 2020 |
| | 22 September to 21 October, 2015 | 3.1 | Beijing | Wang et al., 2017a |

10. *A number of correlations are explored between P-unknown and various other data metrics (e.g., trace gas concentrations, light intensity, PM$_{2.5}$ concentrations, and products thereof). A number of correlations are reported using R values as an indicator of the quality of the fit. However, it is unclear whether these correlations are statistically significant. Please provide information on statistical significance. Also, with respect to the correlations, I am uncomfortable with choosing only the months that support a given hypothesis. For example, it was noted that there is a strong correlation (R = 0.62) between P-unknown and (JNO$_2$ x NO$_2$ x PM$_{2.5}$) in June, although this is the only month where this correlation seems to be significant. Yet, this is taken to be evidence for a light-induced heterogeneous reaction for NO$_2$-to-HONO conversion. Why would this relationship only exist in June and not during other months. Same for the correlations with various salt concentrations in October (lines 400-405).*

Thank you for this suggestion. The statistical significance of the correlations (P value) was added in Table S4 and in the main manuscript. Indeed, making an assumption based on the data from just a

few months would be somewhat hasty. Therefore, we have added qualifiers such as "in this observation" to the corresponding inferences.

11. *Section 3.5: This section explores the relationships between HONO concentrations, $PM_{2.5}$ and ozone concentrations in the dataset. A positive correlation between particle pollution and HONO concentration in summer was taken to be evidence that particles are the source of HONO. However, correlation does not imply causation and it is possible that both $PM_{2.5}$ and HONO are stem from the same sources (i.e., their concentrations would both increase during pollution events) and it is also possible that high HONO concentrations can lead to higher oxidative capacity and therefore higher rates of aerosol formation.*

Thank you for this comment. According to the suggestions above, the explanations about the relationships between HONO and $PM_{2.5}$ concentrations have been updated.

(Page 21, line 434-438). "One possible explanation of this phenomenon was that the increase in particle pollution in summer and autumn might lead to the formation of HONO and an increase in its concentration. Another possible explanation was that high HONO concentrations could lead to higher oxidative capacity and therefore higher rates of aerosol formation. There was also a possible explanation as both $PM_{2.5}$ and HONO were stem from the same sources (i.e., their concentrations would both increase during pollution events)."

12. *Supporting Information figures and tables: Place each figure or table on its own page and ensure that the figure captions are on the same page as the graphs or tables.*

This has been corrected in Supporting Information. All figures and tables are placed within a single page width, and the captions are on the same page as the graphs or tables.

13. *Figure S1: What does the symbol WD and WS stand for. Please define.*

"WD" is wind direction and "WS" is wind speed, these have been added in Figure S2. (SI, page 8)

14. *Lastly, although I felt the language used in the manuscript was relatively clear to understand, it would benefit from proofreading/editing by a native English speaker.*

Thank you for this suggestion. We will take your suggestions and enhance our English writing skills in our subsequent work.

---

## Author Response (AR2)

*Li et al. present warm season measurements of HONO and related trace gases at a measurement site in Beijing, China. According to these authors, these data and analysis fill a measurement gap for the seasonal understanding of HONO in Beijing. As such, the data and analysis are a reasonable contribution to the literature on this subject and the related topics of the contribution of HONO to free radicals, oxidation capacity, and urban air pollution. It can be published in ACP subject to the comments below.*

*As with previous literature on this topic, the authors find a large missing or unknown source of HONO. The authors analyze the magnitude of this source, and try to assess the mechanisms that may produce it. This is the weakest part of this paper and one that should undergo major revision. The presented correlations do not appear to support any of the proposed mechanisms. Rather than trying to provide evidence for these sources, the authors would do better to simply evaluate the magnitude of each based on the available data and assess the extent to which each can explain the observations. There do not appear to be meaningful correlations that would support any given explanation for the unknown HONO source, but an analysis of the size of each would still be informative. See specific comments below for recommendations.*

*There are several other major revisions required – see the specific comments below. Of particular importance are the explicit recognition of vertical gradients in HONO, and the related topic of the contribution of HONO to the OH budget.*

*This paper has already been reviewed by two other reviewers. One recommended rejection on the basis of $NO_2$ measured by a molybdenum converter. The authors have addressed this concern, although they could go further with an uncertainty budget for the corrected NOx and all measurements. See specific comments. The second reviewer recommended major revisions based on lack of clarity and consistency in the analyses. I concur with this reviewer, and my recommendations are similar to those of this reviewer. The authors have addressed some of these comments, but have not fully addressed them. See specific comments below.*

Response: We are very grateful to Anonymous Referee #3 for reviewing this manuscript so carefully. We have tried our best to improve and made some changes in the manuscript. We have responded to the comments below in blue text. The revisions in the manuscript are highlighted in yellow color. The response and changes are listed below.

*Specific comments:*

1. *Line 15: Define the emission factor in the abstract – i.e., relative to NOx or some other component of vehicle emissions.*
   Thank you for this suggestion. We have refined this definition (Page 1, line 16).

2. *Line 17: Conversion frequency in the abstract appears to be a first order rate constant. If so, quote as such, and don't give in units of % per hour but rather in $s^{-1}$. Also check this unit as it seems quite slow as given.*
   Thank you for this suggestion. We have refined the expression by removing the percentage and changing the range values to the average ($0.008\ h^{-1}$). The averaged value is within a reasonable range comparing with Xuan et al. (2023) and Jia et al. (2020), $0.0073\ h^{-1}$ and $0.0078\ h^{-1}$ from August to September, 2018 (Page 1, line 17).
   References:

Xuan, H., Zhao, Y., Ma, Q., Chen, T., Liu, J., Wang, Y., Liu, C., Wang, Y., Liu, Y., Mu, Y., and He, H.: Formation mechanisms and atmospheric implications of summertime nitrous acid (HONO) during clean, ozone pollution and double high-level $PM_{2.5}$ and $O_3$ pollution periods in Beijing, Sci. Total Environ., 857, 159538, doi: 10.1016/j.scitotenv.2022.159538, 2023.

Jia, C. H., Tong, S. R., Zhang, W. Q., Zhang, X. R., Li, W. R., Wang, Z., Wang, L. L., Liu, Z. R., Hu, B., Zhao, P. S., and Ge, M. F.: Pollution characteristics and potential sources of nitrous acid (HONO) in early autumn 2018 of Beijing, Sci. Total Environ., 735, 11, doi: 10.1016/j.scitotenv.2020.139317, 2020.

3. *Line 50-52: OH + NO is not normally a net source of HONO as it is balanced by photolysis to create a null cycle that does not affect OH or NOx.*

   The content of the introduction mainly summarizes the source-sink characteristics of HONO. OH+NO is an important source of HONO, although it is not a net increase of HOx. We have provided a more detailed explanation in the manuscript. (Page 2, line 43-48)

   "Homogeneous reaction of OH +NO. This is an important source of HONO. Although the reaction of OH+NO, which is the reverse reaction of HONO photolysis, does not contribute to an actual increase in free radicals, the assessment of this reaction pathway is significant for understanding the sources and sinks of HONO. Especially during the winter pollution period in North China Plain, where there is usually a higher concentration of NO, this reaction pathway will contribute to a higher concentration of HONO (Xue et al., 2020).

   Reference:

   Xue, C., Zhang, C., Ye, C., Liu, P., Catoire, V., Krysztofiak, G., Chen, H., Ren, Y., Zhao, X., Wang, J., Zhang, F., Zhang, C., Zhang, J., An, J., Wang, T., Chen, J., Kleffmann, J., Mellouki, A., and Mu, Y.: HONO Budget and Its Role in Nitrate Formation in the Rural North China Plain, Environmental Science & Technology, 54, 11048-11057, 10.1021/acs.est.0c01832, 2020.

4. *Line 94: $NO_2$ hydrolysis is not the only potential interference in LOPAP instruments for HONO. For example, $HO_2NO_2$ is known to interfere with HONO in these systems. Can the authors provide more detail on the phrase "subtracted by a deployed dual channel absorption system" and explain how this corrects for interferences in the LOPAP method?*

   Briefly, the main structure of the instrument sampling unit is a double-channel stripping coil. In the first coil, almost all of the HONO and a small fraction of interfering species (e.g., $NO_2$, peroxyacetyl nitrate, $NO_2^-$) are absorbed by deionized water; while in the second channel, only a small fraction of interfering species is absorbed, which could be seen as the comparable conversion ratio in both the first and second channels. Therefore, the HONO concentration output by the instrument is the difference in concentration between the first and second channels. As to the soluble species such as $HO_2NO_2$, considering its little ambient concentration, especially in warm season, the interfering could be neglected. And we have added this explanation in the manuscript. (Page 3-4, line 89-95)

   Reference:

   Veres, P. R., Roberts, J. M., Wild, R. J., Edwards, P. M., Brown, S. S., Bates, T. S., Quinn, P. K., Johnson, J. E., Zamora, R. J., and de Gouw, J.: Peroxynitric acid ($HO_2NO_2$) measurements during the UBWOS 2013 and 2014 studies using iodide ion chemical ionization mass spectrometry, Atmospheric Chemistry and Physics, 15, 8101-8114, 10.5194/acp-15-8101-2015, 2015.

5. *Line 102: Does this imply an uncertainty in the analysis that is greater below 7 ppb of NOx? Is an uncertainty budget given? What, in general, is the uncertainty of all the measurements quoted in this paragraph?*

In the comparative observation of two devices, when the $NO_2$ concentration is less than 7ppb, the fitting results show that the ratio is magnified or reduced by 1.14 times. In terms of uncertainty, the fitting $R^2$ value is 0.96. Therefore, for the situation where the $NO_2$ concentration is less than 7ppb, the uncertainty should consider an additional 4% on top of the original 20% measurement uncertainty of the instrument (Yang et al., 2021). Based on the error calculation equation, the overall uncertainty is 20.4%.

Reference:

Yang, X., Lu, K., Ma, X., Liu, Y., Wang, H., Hu, R., Li, X., Lou, S., Chen, S., Dong, H., Wang, F., Wang, Y., Zhang, G., Li, S., Yang, S., Yang, Y., Kuang, C., Tan, Z., Chen, X., Qiu, P., Zeng, L., Xie, P., and Zhang, Y.: Observations and modeling of OH and $HO_2$ radicals in Chengdu, China in summer 2019, Science of the Total Environment, 772, 144829, 10.1016/j.scitotenv.2020.144829, 2021.

6. *Line 117: What is the uncertainty associated with estimating OH via equation 1? See comments above re: uncertainty budget.*

Equation 1 in the manuscript is cited from Liu et al. (2019). The values of coefficients a, b, and c in this equation were adopted from the OH studies in the Pearl River delta (PRD) and Beijing, China (Rohrer et al., 2014; Tan et al., 2017, 2018). The influence of the uncertainty of the coefficients was estimated, results showed that the errors of OH increased with the increase of $J(O^1D)$, but the ratios of error to mean value of OH radicals were in an acceptable range of 0.37-0.55. In other words, the OH radical concentration calculated through this formula was within a reasonable range and would not subvert the relative conclusions in this study. We have added the explanation in the manuscript. (Page 4-5, line 120-122, line 125-126)

References:

Liu, Y., Nie, W., Xu, Z., Wang, T., Wang, R., Li, Y., Wang, L., Chi, X., and Ding, A.: Semi-quantitative understanding of source contribution to nitrous acid (HONO) based on 1 year of continuous observation at the SORPES station in eastern China, Atmos. Chem. Phys., 19, 13289–13308, doi: 10.5194/acp-19-13289-2019, 2019.

Rohrer, F., Lu, K., Hofzumahaus, A., Bohn, B., Brauers, T., Chang, C.-C., Fuchs, H., Häseler, R., Holland, F., Hu, M., Kita, K., Kondo, Y., Li, X., Lou, S., Oebel, A., Shao, M., Zeng, L., Zhu, T., Zhang, Y., and Wahner, A.: Maximum efficiency in the hydroxyl radical- based self-cleansing of the troposphere, Nat. Geosci., 7, 559–563, https://doi.org/10.1038/ngeo2199, 2014.

Tan, Z., Fuchs, H., Lu, K., Hofzumahaus, A., Bohn, B., Broch, S., Dong, H., Gomm, S., Häseler, R., He, L., Holland, F., Li, X., Liu, Y., Lu, S., Rohrer, F., Shao, M., Wang, B., Wang, M., Wu, Y., Zeng, L., Zhang, Y., Wahner, A., and Zhang, Y.: Radical chemistry at a rural site (Wangdu) in the North China Plain: observation and model calculations of OH, $HO_2$ and $RO_2$ radicals, Atmos. Chem. Phys., 17, 663–690, https://doi.org/10.5194/acp-17-663-2017, 2017.

Tan, Z., Rohrer, F., Lu, K., Ma, X., Bohn, B., Broch, S., Dong, H., Fuchs, H., Gkatzelis, G. I., Hofzumahaus, A., Holland, F., Li, X., Liu, Y., Liu, Y., Novelli, A., Shao, M., Wang, H., Wu, Y., Zeng, L., Hu, M., Kiendler-Scharr, A., Wahner, A., and Zhang, Y.: Wintertime photochemistry in Beijing: observations of ROx radical concentrations in the North China Plain during the BEST-ONE campaign, Atmos. Chem. Phys., 18, 12391–12411, https://doi.org/10.5194/acp-18-12391-2018, 2018.

7. *Line 123: Is there a reference to the nighttime OH at 2e-5? This is quite a high number. Does this matter for the subsequent analysis? What is the source of nighttime OH at this level? A sustained OH of this magnitude would require a large, non-photolytic source. If not important to the*

*subsequent analysis, suggest neglecting nighttime OH.*

The nighttime OH radicals mostly come from the ozonolysis of alkenes (Ren et al., 2013; Tan et al., 2019). However, due to limitations in measuring methods, there is a lack of relevant data. The value of 2e-5 is calculated based on the formula. According to the comments, we have removed the subsequent analysis related to nighttime OH radicals.

Reference:

Ren, X., van Duin, D., Cazorla, M., Chen, S., Mao, J., Zhang, L., Brune, W. H., Flynn, J. H., Grossberg, N., Lefer, B. L., Rappenglück, B., Wong, K. W., Tsai, C., Stutz, J., Dibb, J. E., Thomas Jobson, B., Luke, W. T., and Kelley, P.: Atmospheric oxidation chemistry and ozone production: Results from SHARP 2009 in Houston, Texas, Journal of Geophysical Research: Atmospheres, 118, 5770-5780, 10.1002/jgrd.50342, 2013.

Tan, Z., Lu, K., Jiang, M., Su, R., Wang, H., Lou, S., Fu, Q., Zhai, C., Tan, Q., Yue, D., Chen, D., Wang, Z., Xie, S., Zeng, L., and Zhang, Y.: Daytime atmospheric oxidation capacity in four Chinese megacities during the photochemically polluted season: a case study based on box model simulation, Atmospheric Chemistry and Physics, 19, 3493-3513, 10.5194/acp-19-3493-2019, 2019.

8. *Line 171-174: The premise is hard to follow here – why should HONO sources remain constant? HONO should follow NOx (stated earlier), so variation in NOx should lead to variation in HONO.*

Thank you for this comment. We have removed this part of the content.

9. *Line 211: Provide more justification for the use of 2 μg m$^{-3}$ to exclude biomass burning. For example, what is the ratio of HONO/K$^+$ that would allow assessment of biomass burning as a source of HONO?*

According to studies on the influence of biomass burning on HONO chemistry (Nie et al., 2015), when K$^+$ concentration is higher than 2 μg m$^{-3}$ and the ratio of K$^+$ to $PM_{2.5}$ is larger than 0.02, the plumes are defined as biomass burning samples. While the samples with K$^+$ concentrations lower than 2 μg m$^{-3}$ and a ratio of K$^+$ to $PM_{2.5}$ smaller than 0.02 are categorized as non-biomass burning samples. We have added this explanation in the manuscript. (Page 10, line 211-214)

Reference:

Nie, W., Ding, A. J., Xie, Y. N., Xu, Z., Mao, H., Kerminen, V. M., Zheng, L. F., Qi, X. M., Huang, X., Yang, X. Q., Sun, J. N., Herrmann, E., Petäjä, T., Kulmala, M., and Fu, C. B.: Influence of biomass burning plumes on HONO chemistry in eastern China, Atmos. Chem. Phys., 15, 1147–1159, doi: 10.5194/acp-15-1147-2015, 2015.

10. *Line 251: The OH concentration at night is taken from literature and not observed. Is this level consistent with the observed NO and NO$_2$? Such high levels of NOx would have the effect of greatly reducing nighttime OH, so the literature values would also have to have similar NOx levels. The high inferred levels of OH are not plausible without also demonstrating that there is a large OH source. Absent such an analysis, the nighttime HONO source from OH + NO should be omitted. It is almost certainly the case that the quoted values are an upper limit, perhaps a large upper limit, to the actual contribution of this reaction at night.*

Following the suggestions, we have removed all contents related to the OH radicals at nighttime.

11. *Line 308-310: Vertical gradients in HONO are well known (e.g., multiple references from Stutz et al., see below, Vanden Boer et al. 2013). Why would vertical transport be negligible?*

We have carefully read the literatures. And according to literatures mentioned above (Wong et al., 2011; Wong et al., 2012; Wong et al., 2013; Pinto et al., 2014; Stutz et al., 2002; Wang et al., 2006; VandenBoer et al., 2013; Young et al., 2012), photolytic HONO formation at the ground is the major formation pathway in the lowest 20 m, while a combination of gas-phase, photolytic formation on

aerosol, and vertical transport is responsible for daytime HONO between 200-300 meters above the ground. In our work, the measurement was conducted on the rooftop of one building, about eight meters above the ground. Therefore, the contribution of vertical transport to the near-surface HONO source is not significant. We have revised the statement in the manuscript and added the related literatures to the manuscript. (Page 12-13, line 283-288)

References

Wong, K.W., H.J. Oh, B.L. Lefer, B. Rappenglück, and J. Stutz, Vertical profiles of nitrous acid in the nocturnal urban atmosphere of Houston, TX. Atmos. Chem. Phys., 2011. 11(8): p. 3595-3609.

Wong, K.W., C. Tsai, B. Lefer, C. Haman, N. Grossberg, W.H. Brune, X. Ren, W. Luke, and J. Stutz, Daytime HONO Vertical Gradients during SHARP 2009 in Houston, TX. Atmos. Chem. Phys., 2012. 12: p. 635-652.

Wong, K.W., C. Tsai, B. Lefer, N. Grossberg, and J. Stutz, Modeling of daytime HONO vertical gradients during SHARP 2009. Atmos. Chem. Phys., 2013. 13(7): p. 3587-3601.

Pinto, J.P., J. Dibb, B.H. Lee, B. Rappenglück, E.C. Wood, M. Levy, R.Y. Zhang, B. Lefer, X.R. Ren, J. Stutz, C. Tsai, L. Ackermann, J. Golovko, S.C. Herndon, M. Oakes, Q.Y. Meng, J.W. Munger, M. Zahniser, and J. Zheng, Intercomparison of Field Measurements of Nitrous Acid (HONO) during the SHARP Campaign. Journal of Geophysical Research: Atmospheres, 2014. p. 2013JD020287.

Stutz, J., B. Alicke, and A. Neftel, Nitrous acid formation in the urban atmosphere: Gradient measurements of $NO_2$ and HONO over grass in Milan, Italy. Journal of Geophysical Research-Atmospheres, 2002. 107(D22).

Wang, S., R. Ackermann, and J. Stutz, Vertical profiles of NOx chemistry in the polluted nocturnal boundary layer in Phoenix, AZ: I. Field observations by long-path DOAS. Atmos. Chem. Phys., 2006. 6: p. 2671-2693.

VandenBoer, T.C., S.S. Brown, J.G. Murphy, W.C. Keene, C.J. Young, A.A.P. Pszenny, S. Kim, C. Warneke, J. de Gouw, J.R. Maben, N.L. Wagner, T.P. Riedel, J.A. Thornton, D.E. Wolfe, W.P. Dubé, F. Öztürk, C.A. Brock, N. Grossberg, B. Lefer, B.M. Lerner, A.M. Middlebrook, and J.M. Roberts, Understanding the role of the ground surface in HONO vertical structure: High resolution vertial profiles during NACHTT-11. J. Geophys. Res., 2013. 118(17): p. 10155-10171.

Young, C.J., R.A. Washenfelder, L.H. Mielke, H.D. Osthoff, P. Veres, A.K. Cochran, T.C. VandenBoer, H. Stark, J. Flynn, N. Grossberg, C.L. Haman, B. Lefer, J.B. Gilman, W.C. Kuster, C. Tsai, O. Pikelnaya, J. Stutz, J.M. Roberts, and S.S. Brown, Vertically resolved measurements of nighttime radical reservoirs in Los Angeles and their contribution to the urban radical budget. Environ. Sci. Technol., 2012. 46: p. 10965-10973.

12. *Line 366-367: If the phenomenon is not evident in four of the five months, then it is clear that there is not evidence for it. The conclusions should clearly state this finding rather than implying that it operates in June only.*

Thank you for this comment. We have modified the corresponding descriptions. (Page 16, line 342-346)

"In our work, good correlation between $P_{unknown}$ and product of $JNO_2$ and RH was found in June. However, this phenomenon was not evident in other four months. The phenomenon was not evident in four of the five months, this showed that in this observation there was not strong evidence for this conclusion. As shown in Table S4, June had the lowest RH and the highest $JNO_2$ value, the other four months had relative higher RH (due to the precipitation) and lower $JNO_2$ value. This phenomenon may be closely related to meteorological conditions and requires further research for

validation"

13. *Line 371-374: The analysis is very confusing. There is a correlation of the unknown source with the product of jNO$_2$, NO$_2$ and PM$_{2.5}$ that is evident only in June? If so, there is no evidence for such a source, and the conclusion should state this. Additionally, none of the R values (which all produce r$^2$ well under 0.5) are convincing. The assumed variations account for far less than half of the observed variability (r$^2$ values all well smaller than 0.5).*

    Thank you for this comment. We have removed this section of the analysis.

14. *Line 375-381: Similar comment to above. The correlations presented offer no evidence for the source being tested. One could instead simply calculate the magnitude of the source based on previous literature and compare this to the observed P$_{unknown}$. The observations themselves appear to provide no evidence.*

    Thank you for this comment. We have removed this section of the analysis.

15. *Line 389: Same comment. Based on the correlation coefficients presented, the conclusion should be that there is no evidence for this source, but that it could contribute based on prior literature.*

    Thank you for this comment. We have made modifications to the corresponding explanations. (Page 17, line 364-365)

    "However, the phenomenon above were not evident in four of the five months, this showed that in this observation there was not strong evidence for these conclusions, but that they could contribute based on prior literatures"

16. *Line 407-409: The conclusion is flawed, since the source should not vary from month to month without an obvious mechanism. A more likely explanation is simple variation in the data, which can simply be presented as an average and standard deviation rather than as a time varying source. This comment was prominent in previous reviews and should be addressed.*

    Thank you for this suggestion. We have removed this section of the analysis.

17. *Section 3.4.3: There is an important caveat missing in this section in that it pertains to the OH source at the altitude of the measurement. Since vertical gradients in HONO are well known (see above) but not measured here, the analysis must be specified as a local analysis at a fixed height rather than characteristic of the entire mixed layer. The actual contribution to OH is smaller, and likely much smaller, than shown here when integrated across the mixed layer.*

    Thank you for this comment. We have added the corresponding limiting conditions in the manuscript. (Page 17, line 367-368)

    "the OH production rate from HONO (P$_{OH-HONO}$) at the CRAES observation site was calculated in this work"

18. *Also in this section, there is no comparison to the photolysis of formaldehyde, a large and known HOx source in urban areas. Presumably this is due to the lack of formaldehyde measurements. If so, this should be clearly stated, and it should also be stated that the analysis is not a full HOx buddget.*

    Thank you for this suggestion. Indeed, we did not measure formaldehyde in this work, and we have added the explanations in the manuscript. (Page 17, line 368-369)

    "As the formaldehyde was not measured in this work, which was a large and known HO$_x$ source in urban areas, thus the analysis here was not a full HO$_x$ buddget."

19. *Lines 418-420: What is P$_{OH-O3}$? Is this O$_3$ photolysis to O$_{1D}$ followed by reaction with water vapor?*

    P$_{OH-O3}$ is the production rate of OH radical via O$_3$ photolysis to O1D followed by reaction with water vapor. And we have added the explanation in the manuscript. (Page 17, line 371-372)

---

## Author Response (AR4)

*The authors have satisfactorily addressed most of my comments. There is only one outstanding comment that should be addressed prior to publication.*

*Reviewer Comment 11: Vertical gradients in HONO are well known (e.g., multiple references from Stutz et al., see below, VandenBoer et al. 2013). Why would vertical transport be negligible?*

*Comment 11 Author Response: We have carefully read the literatures. And according to literatures mentioned above (Wong et al.,2011; Wong et al., 2012; Wong et al., 2013; Pinto et al., 2014; Stutz et al., 2002; Wang et al., 2006;VandenBoer et al., 2013; Young et al., 2012), photolytic HONO formation at the ground is the major formation pathway in the lowest 20 m, while a combination of gas-phase, photolytic formation on aerosol, and vertical transport is responsible for daytime HONO between 200-300 meters above the ground. In our work, the measurement was conducted on the rooftop of one building, about eight meters above the ground. Therefore, the contribution of vertical transport to the near-surface HONO source is not significant. We have revised the statement in the manuscript and added the related literatures to the manuscript. (Page 12-13, line 283-288).*

*The response does not fully address the original comment. It would not be possible to conclude that vertical transport is unimportant without also calculating the rate of vertical exchange of the air mass for an 8 m height in which vertical gradients are known to exist. This point must be addressed prior to publication. If it is not possible to calculate a vertical exchange rate, then the statement that vertical transport is negligible must be removed. In its place, a statement is required that vertical exchange is unknown since the data to calculate its effect on HONO measured at 8 m is not available. Therefore, all subsequent analysis relies on the assumption that vertical exchange is unimportant, but this assumption represents an uncertainty that is not easily quantified.*

Response: We thank Anonymous Referee #3 for the comment. Indeed, due to the unavailability of corresponding data, we are unable to calculate the vertical exchange rate. We have made revisions in the main text according to the reviewer's advice.

"According to related observations (Wong et al., 2011; Wong et al., 2012; Wong et al., 2013; Pinto et al., 2014; Stutz et al., 2002; Wang et al., 2006; VandenBoer et al., 2013; Young et al., 2012), photolytic HONO formation at the ground is the major formation pathway in the lowest 20 m, while a combination of gas-phase, photolytic formation on aerosol, and vertical transport is responsible for daytime HONO between 200-300 meters above the ground. In our work, the measurement was conducted on the rooftop of one building, about eight meters above the ground. As vertical exchange is unknown since the data to calculate its effect on HONO measured at 8 m is not available. Therefore, all subsequent analysis relies on the assumption that vertical exchange is unimportant, but this assumption represents an uncertainty that is not easily quantified" (Page 12-13, line 284-291)